# Spatio-temporal dynamics of three diseases caused by *Aedes*-borne arboviruses in Mexico

Bo Dong[1], Latifur Khan[1], Madison Smith [2], Jesus Trevino [3], Bingxin Zhao[4], Gabriel L. Hamer[5], Uriel A. Lopez-Lemus[6], Aracely Angulo Molina [7], Jailos Lubinda[8], Uyen-Sa D. T. Nguyen [2] & Ubydul Haque[2✉]

**Abstract**

**Background** The intensity of transmission of *Aedes*-borne viruses is heterogeneous, and multiple factors can contribute to variation at small spatial scales. Illuminating drivers of heterogeneity in prevalence over time and space would provide information for public health authorities. The objective of this study is to detect the spatiotemporal clusters and determine the risk factors of three major *Aedes*-borne diseases, Chikungunya virus (CHIKV), Dengue virus (DENV), and Zika virus (ZIKV) clusters in Mexico.

**Methods** We present an integrated analysis of *Aedes*-borne diseases (ABDs), the local climate, and the socio-demographic profiles of 2469 municipalities in Mexico. We used SaTScan to detect spatial clusters and utilize the Pearson correlation coefficient, Randomized Dependence Coefficient, and SHapley Additive exPlanations to analyze the influence of socio-demographic and climatic factors on the prevalence of ABDs. We also compare six machine learning techniques, including XGBoost, decision tree, Support Vector Machine with Radial Basis Function kernel, K nearest neighbors, random forest, and neural network to predict risk factors of ABDs clusters.

**Results** DENV is the most prevalent of the three diseases throughout Mexico, with nearly 60.6% of the municipalities reported having DENV cases. For some spatiotemporal clusters, the influence of socio-economic attributes is larger than the influence of climate attributes for predicting the prevalence of ABDs. XGBoost performs the best in terms of precision-measure for ABDs prevalence.

**Conclusions** Both socio-demographic and climatic factors influence ABDs transmission in different regions of Mexico. Future studies should build predictive models supporting early warning systems to anticipate the time and location of ABDs outbreaks and determine the stand-alone influence of individual risk factors and establish causal mechanisms.

**Plain language summary**

The rate of the spread of diseases caused by the Chikungunya, Dengue, and Zika viruses varies in space and time. Here, we aimed to identify the causes of such variation in the population with the disease in a given time period and specific area. To identify some of these factors we analyzed local climate and socio-demographic profiles of 2469 municipalities in Mexico and how these related to the presence of the diseases caused by Chikungunya, Dengue, and Zika viruses. We detected that the areas with most cases of these diseases at a certain time were influenced both by socio-demographic and climatic factors, but socio-economic factors are more influential in predicting the outbreaks. This information could help health authorities predict outbreaks and plan better how to target them.

[1] Department of Computer Science, University of Texas at Dallas, Richardson, TX 75080, USA. [2] Department of Biostatistics and Epidemiology, University of North Texas Health Science Center, Fort Worth, TX, USA. [3] Department of Urban Affairs at the School of Architecture, Universidad Autónoma de Nuevo León, 66455 San Nicolás de los Garza, Nuevo Léon, Mexico. [4] Department of Statistics and Data Science, University of Pennsylvania, Philadelphia, PA 19104, USA. [5] Department of Entomology, Texas A&M University, College Station, TX, USA. [6] Department of Health Sciences, Center for Biodefense and Global Infectious Diseases, Colima 28078, Mexico. [7] Department of Chemical and Biological Sciences, University of Sonora, Hermosillo 83000 Sonora, Mexico. [8] Telethon Kids Institute, Malaria Atlas Project, Nedlands, WA, Australia. ✉email: mdubydul.haque@unthsc.edu

The three most important viruses transmitted by *Aedes aegypti* mosquitoes include chikungunya (CHIKV), dengue virus (DENV), and Zika virus (ZIKV)[1]. These *Aedes*-borne diseases (ABDs) are considered human-amplified urban arboviruses because humans play the primary reservoir, facilitating virus amplification. The exact global burden of CHIKV and ZIKV is unknown. The prevalence of DENV has increased dramatically worldwide in recent decades. Over the last two decades, the number of DENV cases increased over eightfold[2]. Nearly 105 million DENV infections are reported globally per year[3], with ~51 million febrile DENV cases and four million symptomatic infections requiring hospitalization[3]. The Latin American countries alone had an estimated 16% of the global DENV burden[3]. These three ABDs are all common in Mexico and are reported within 57% of its municipalities[4,5]. In 2019, Mexico had one of the highest numbers of reported DENV cases in Latin America, along with Brazil, which had a slightly higher number of reported cases[2].

Due to its continuously immense burden on public health, it is crucial to integrate arbovirus control efforts[6]. For this purpose, spatiotemporal cluster detection techniques can serve as a useful research tool[7,8]. Temperature and rainfall have both positive and negative effects on ABDs outbreaks[9]. Differences in the landscape, climatic variations, and socio-economic development result in differences in transmission potential among locations. Climatic parameters such as rainfall, wind speed, and temperature are important drivers in mosquito development and virus reproduction[10]. In addition, socio-economic factors such as barriers to healthcare services, inadequate sanitation, poverty, living in a poor neighborhood, and poor water supply were associated with the transmission of ABDs[11–15].

Specific risk factors associated with CHIKV, DENV, and ZIKV are also multifactorial. The risk of exposure to DENV is influenced by rainfall, temperature, relative humidity, and unplanned rapid urbanization[16,17]. Smaller to larger DENV outbreaks were associated with increased temperature (23.8–33.1 °C) and the delayed effects could be predicted with a one-week lag[18]. Mean (>27 °C), minimum (>22 °C), and maximum temperature (>38 °C) were found to be the most favorable weather condition at a lag of 1–3 months in the tropical and subtropical climate zone, respectively[19,20]. Monthly mean rainfall showed a positive correlation with monthly DENV cases at a lag of 1–3 months[19–21]. An increase of 1 mm of rainfall with a lag of 2–3 weeks was associated with 1.3–2.1% more DENV cases[22]. Another study also suggested that an increase of 1% in rainfall corresponded to an increase of 3.3% in the DENV cases[21]. ZIKV prevalence was greater in neighborhoods with little access to municipal water infrastructure, whereas CHIKV prevalence was weakly correlated with urbanization[23]. In addition, another study determined both total rainfall and average temperature were the best meteorological factors to predict ZIKV infection[24].

While certain studies conducted in Mexico revealed that specific climate factors are strong drivers for these ABDs[25–27], other studies suggested socioeconomic factors were the strongest predictors in the spread of arthropod-borne (or arbovirus) transmission in some parts of Mexico[28–30]. Despite Mexico being a country highly endemic to CHIKV, DENV, and ZIKV[4,5,31,32], there is no national study to date in which long-time series data have been combined with spatiotemporal socio-demographic risk factors and climatic parameters. Understanding the spatiotemporal and socio-demographic risk factors (e.g. access to improved water, housing quality, population density, sanitation) and climatic factors (e.g. temperature and rainfall) associated with the risk of these three ABDs is key for informing vector control programs and predicting the time and location of outbreaks.

The hypothesis of this study is that there are geographic clusters of CHIKV, DENV, and ZIKV in Mexico. The geographic clusters can be associated with either socioeconomics or climatic parameters. For some clusters, the multifactorial causes (e.g., socio-economic features may have more impact on the prevalence of ABDs, whereas, in other clusters, climatic features have the greatest impact) can be isolated to determine the stand-alone influence of individual risk factors. To the best of our knowledge, no such analysis has ever been performed in Mexico. Therefore, to address this gap, this study aims to detect the spatiotemporal clusters and determine the risk factors of CHIKV, DENV, and ZIKV clusters in Mexico with lab-confirmed human cases from 2012 to 2019 using machine learning approaches.

## Method

**Study area.** Mexico, the southernmost country in North America, has 32 states, 2469 municipalities, and an estimated population of 126 million[33]. With its high population density and diverse weather conditions (including tropical zones), Mexico has an ideal environment for vector-borne diseases. Northern Mexico has an arid climate characterized by hot summers and sporadic rainfall. In contrast, southern Mexico observes more than 2000 mm of rainfall annually (Fig. S1)[33]. Although vastly different, both regions facilitate optimal conditions for vector-borne diseases, including ABDs[34,35].

**Disease prevalence data.** The dataset was compiled from the daily reported individual-level data for CHIKV, DENV, and ZIKV. To collect information for this dataset, state public health laboratories of Mexico began by identifying cases of CHIKV, DENV, and ZIKV. Confirmed cases were reported to the local health facility within 24 h of detection. These cases were relayed to the General Directorate of Epidemiology, which gathers national data[36]. After gathering data from the General Directorate of Epidemiology, we assessed de-identified daily case records of Mexico's national data of arboviral disease (or arbovirus infection). This includes information from 2469 municipalities over the period between January 2012 and December 2019.

**Spatial data.** We used the Geographic Information Systems (GIS) package, ArcGIS version 10.7 (Environmental Systems Resource Institute; [ESRI], Redlands, CA), to create municipality-based shapefile centroids in the UTM projection system to which the recorded surveillance data was appended. Altitude was calculated based on the municipality center.

**Climate data.** Monthly temperature data, measured as surface air temperature at 2-m height, were obtained for municipalities from the Climate Forecast System Reanalysis (CFSR) dataset of the National Centers for Environmental Prediction (NCEP)[37]. Monthly precipitation data were obtained for each municipality from the Climate Hazards Group Infrared Rainfall with Stations (CHIRPS) dataset[38]. We prepared the daily average climatic parameters (rainfall and temperature) in Mexico throughout the 8-year study period. We also used the daily mean, minimum, and maximum of temperatures as well as the daily mean, minimum, and maximum of rainfall (mm) as the primary climate parameters (Fig. S1). All climate variables were obtained for the period from 2012 to 2019.

**Population, entomology, rural/ urban, and socio-economic data.** For each municipality, the Mexican National Council carried out the collection of socio-economic data and the calculation of average change for Evaluating the Social Development Policy (Consejo Nacional de Evaluación de la Política de Desarrollo Social

or, CONEVAL) using the national census data[39]. We used illiteracy, populations without health services, houses with dirt floors, houses without a toilet facility, houses without water pipelines, houses without a sewage system, and houses without electricity as the socio-economic parameters (Supplementary Fig. S2). Based on socio-economic variables in 2005 and 2015, we built a time series ARIMA model to project the socio-economic variables from 2012 to 2019. We extracted population density and rural/ urban classification for each municipality from Consejo Nacional de Población[40]. A population size of <10,000 per municipality was considered rural, and >10,000 was considered urban[41]. Presence points of *Ae. aegypti* and *Ae. albopictus* at the municipality level was compiled from 1993 to 2016 across all municipalities in Mexico. This entomological dataset was collected and reported based on the Mexican national vector surveillance guidelines[42].

**Statistics and reproducibility**. SaTScan (v. 9.6.1) was used to detect spatial clusters separately for CHIKV, DENV, and ZIKV (settings: spatial analysis; discrete Poisson probability model; latitude/longitude coordinates; no geographical overlap; scanning for clusters with high rates). Spatial clusters were determined by calculating the maximum-likelihood ratio. Standardized prevalence ratios were estimated by dividing the number of observed cases by the number of expected cases in each cluster. Simulated *p*-values were obtained using Monte Carlo methods with 9999 replications[43]. For further detail, refer to the supplement section.

*The clustering method and evaluation of the clusters.* Under the null hypothesis, and in absence of covariates, it is expected that the number of ABDs in each municipality is proportional to its population size. The Poisson model requires the total counts of ABDs and population counts in each year and geographical coordinates for each municipality. The goal was to detect the statistically significant geographic clusters and identify the risk factors behind the clusters.

We used the Pearson correlation coefficient[44], randomized dependence coefficient (RDC)[45], and SHapley Additive exPlanations (SHAP)[46] to assess the stand-alone influence of the socio-economic and climate factors on each arbovirus cluster.

**Pearson correlation coefficient**: The Pearson product–moment correlation coefficient (or Pearson correlation coefficient, for short) is a measure of the strength of a linear association between two variables and is denoted by $r$. The Pearson correlation coefficient (PCC) is defined as the covariance of the two variables divided by the product of their standard deviations. A Pearson product–moment correlation attempts to draw a line of best fit through the data of two variables, and the Pearson correlation coefficient, $r$, indicates how far away all these data points are to this line of best fit (i.e., how well the data points fit this new model/line of best fit). The Pearson correlation coefficient, $r$, can take a range of values from $+1$ to $-1$. A value of 0 indicates that there is no association between the two variables. A value $>0$ indicates a positive association; that is, as the value of one variable increases, so does the value of the other variable. A value $<0$ indicates a negative association; that is, as the value of one variable increases, the value of the other variable decreases. The stronger the association of the two variables, the closer the Pearson correlation coefficient, $r$, will be to either $+1$ or $-1$ depending on whether the relationship is positive or negative, respectively.

*The randomized dependence coefficient (RDC).* The randomized dependence coefficient (RDC)[45] is a measure of nonlinear dependence between random variables of arbitrary dimension based on the Hirschfeld–Gebelein–Renyi maximum correlation coefficient. Given the random samples, $X \in R^{p \times n}$ and $Y \in R^{q \times n}$ and the parameters $k \in N_+$ and $s \in R_+$, the randomized dependence coefficient between $X$ and $Y$ is defined as

$$\mathrm{rdc}(X, Y; k, s) := \sup_{\alpha, \beta} \rho(a^{\mathrm{T}} \Phi(P(X); k, s), \beta^{\mathrm{T}} \Phi(P(Y); k, s)) \quad (1)$$

$\Phi(P(X); k, s)$ is a map from $X$ to $\Phi(P(X); k, s)$. $\alpha, \beta$ are pairs of basis vectors such that the projections $a^{\mathrm{T}} X$ and $\beta^{\mathrm{T}} Y$ of two random samples $X \in R^{p \times n}$ and $Y \in R^{q \times n}$ are maximally correlated. RDC is defined in terms of the correlation of random nonlinear copula projections; it is invariant with respect to marginal distribution transformations. RDC is a computationally, efficient, copula-based measure of dependence between multivariate random variables. RDC is invariant with respect to nonlinear scaling of random variables, is capable of discovering a wide range of functional association patterns, and takes a value of zero at independence.

*SHapley Additive exPlanations (SHAP).* SHAP is a game-theoretic approach to explain the output of any machine learning model. It connects optimal credit allocation with local explanations using the classic Shapley values from game theory and their related extensions. SHAP values as a unified measure of feature importance. These are the Shapley values of a conditional expectation function of the original model; thus, they are the solution to the following equation:

$$\phi_i(f, x) = \sum_{z' \in x'} \frac{|z'|!(M - |z'| - 1)!}{M!} [f_x(z') - f_x(z'i)] \quad (2)$$

where $|z'|$ is the number of non-zero entries in $z'$, and $z' \in x'$ represents all $z'$ vectors where the non-zero entries are a subset of the non-zero entries in $x'$. Understanding why a model makes a certain prediction can be as critical as its accuracy in many applications. However, the highest accuracy for large modern datasets is often achieved by complex models that even experts struggle to interpret, such as ensemble or deep learning models, creating a tension between accuracy and interpretability. In response, various methods have recently been proposed to help users interpret the predictions of complex models. However, it is often unclear how these methods are related and when one method is preferable to another.

SHAP assigns each feature an important value for a particular prediction. Its novel components include (1) identifying a new class of additive feature important measures and (2) theoretical results exemplifying a unique solution in this class with a set of desirable properties. The new class unifies six existing methods, of notable importance due to several recent methods lacking the proposed desirable properties in their class. Based on insights from this unification, SHAP demonstrates improved computational performance and/or better consistency with human intuition than previous approaches. In this study, we used SHAP to analyze the impact of model output with respect to different features. In addition, we summarized the impact of socio-economic features, and climate features separately for different clusters.

By using PCC, RDC, and SHAP, we got the impact of each factor simultaneously. For this, we first normalized the weight of socio-economic and climate features for each method separately. For example, for a given method, the importance of socio-economic features is calculated by taking the average importance of these features. We repeated the same approach to calculate climate features. Then, the weight/importance of socio-economic and climate features are calculated in the following way: weight/ importance of socio-economic features = socio-economic attributes impact/(socio-economic attributes impact + climate impact).

**Table 1 Classification performance of various ML algorithms on dengue, zika and CHIKV virus prediction for all clusters.**

| Method | Dengue | | | | | Zika | | | | | CHIKV | | | | |
|---|---|---|---|---|---|---|---|---|---|---|---|---|---|---|---|
| | Accuracy | Weighted accuracy | Precision | Recall | F-score | Accuracy | Weighted accuracy | Precision | Recall | F-score | Accuracy | Weighted accuracy | Precision | Recall | F-score |
| XGBoost | 0.93 | 0.78 | 0.86 | 0.78 | 0.81 | 0.99 | 0.61 | 0.76 | 0.61 | 0.65 | 0.99 | 0.59 | 0.87 | 0.59 | 0.65 |
| Decision tree | 0.91 | 0.78 | 0.77 | 0.78 | 0.78 | 0.98 | 0.63 | 0.63 | 0.63 | 0.63 | 0.98 | 0.63 | 0.63 | 0.63 | 0.63 |
| SVM | 0.90 | 0.60 | 0.86 | 0.60 | 0.64 | 0.99 | 0.50 | 0.50 | 0.50 | 0.50 | 0.99 | 0.50 | 0.50 | 0.50 | 0.50 |
| KNN | 0.90 | 0.73 | 0.77 | 0.73 | 0.75 | 0.99 | 0.52 | 0.62 | 0.52 | 0.54 | 0.99 | 0.55 | 0.70 | 0.55 | 0.58 |
| Random forest | 0.92 | 0.76 | 0.85 | 0.76 | 0.79 | 0.99 | 0.54 | 0.70 | 0.54 | 0.57 | 0.99 | 0.54 | 0.76 | 0.55 | 0.58 |
| Neural network | 0.92 | 0.75 | 0.81 | 0.75 | 0.78 | 0.99 | 0.59 | 0.73 | 0.59 | 0.63 | 0.99 | 0.61 | 0.73 | 0.61 | 0.65 |

**Table 2 Classification performance of various ML algorithms on dengue, zika, and CHIKV virus prediction for non-clusters.**

| Method | Dengue | | | | | Zika | | | | | CHIKV | | | | |
|---|---|---|---|---|---|---|---|---|---|---|---|---|---|---|---|
| | Accuracy | Weighted accuracy | Precision | Recall | F-score | Accuracy | Weighted accuracy | Precision | Recall | F-score | Accuracy | Weighted accuracy | Precision | Recall | F-score |
| XGBoost | 0.98 | 0.77 | 0.89 | 0.77 | 0.82 | 0.99 | 0.53 | 0.85 | 0.53 | 0.56 | 0.99 | 0.62 | 0.72 | 0.62 | 0.65 |
| Decision tree | 0.98 | 0.79 | 0.79 | 0.79 | 0.79 | 0.99 | 0.61 | 0.65 | 0.61 | 0.63 | 0.99 | 0.64 | 0.58 | 0.64 | 0.60 |
| SVM | 0.98 | 0.58 | 0.90 | 0.58 | 0.64 | 0.99 | 0.50 | 0.50 | 0.50 | 0.50 | 0.99 | 0.50 | 0.50 | 0.50 | 0.50 |
| KNN | 0.98 | 0.73 | 0.81 | 0.73 | 0.76 | 0.99 | 0.51 | 0.61 | 0.51 | 0.52 | 0.99 | 0.53 | 0.57 | 0.53 | 0.54 |
| Random forest | 0.98 | 0.75 | 0.87 | 0.75 | 0.79 | 0.99 | 0.54 | 0.74 | 0.54 | 0.57 | 0.99 | 0.54 | 0.72 | 0.54 | 0.56 |
| Neural network | 0.98 | 0.76 | 0.85 | 0.76 | 0.80 | 0.99 | 0.50 | 0.50 | 0.50 | 0.50 | 0.99 | 0.53 | 0.56 | 0.53 | 0.54 |

We separately used CHIKV, DENV, and ZIKV cases as outcome variables in our dataset, and socio-economic variables, population density, urban/rural information, altitude, seasonality, and presence of *Ae. aegypti* and *Ae. albopictus* and climate variables as our features (predictor) to build the model. We computed SHAP values based on the XGBoost model, which showed the important weight for each feature concerning our model. RDC is another approach that can reflect the relationship between features and the target variable. For every feature in our dataset, we computed the important coefficient between this feature and the target variable based on this RDC approach. We computed the correlation coefficient between every feature and the target variable for Pearson coefficients based on the covariance of our dataset.

Next, to summarize/combine the result, we developed two evaluation metrics: majority voting and average. Here, we used three methods to predict the important weight of socio-economic and climate factors. The majority voting metric gives the weighted impact for socio-economic and climate attributes based on the majority of SHAP, RDC, and Pearson results. For example, in the majority voting metric for a specific cluster, if SHAP and RDC values indicated that socio-economic attributes had more impact than climate attributes, we took the average of the SHAP values and the RDC results' values as the majority voting result for this cluster. For the average metric, we took the average result of SHAP, RDC, and Pearson as the average result.

The data distributions may vary based on different ABDs. To ensure we consider each ABD separately and did not introduce data distribution bias, we conducted a stratified analysis to address the potential biases and examined the magnitude and 95% CI in the associations between predictors (independent variables) with CHIKV, DENV, and ZIKV outcomes separately for urban and rural areas.

We also compared six commonly used prediction methods for the best model, such as XGBoost, decision tree, SVM with RBF kernel, KNN (K nearest neighbors) with five neighbors, random forest with six estimators, and neural network with 100 hidden layers. XGBoost is an implementation of gradient-boosted decision trees designed for speed and performance.

*Accuracy, weighted accuracy, precision, recall and F1 scores.* Accuracy measures how often the classifier makes the correct predictions and the ratio between the number of correct predictions and the total number of predictions. However, if the dataset is imbalanced, then the accuracy may not be a good evaluation metric, since here, it only considers the correct predictions and does not care about the instance from which class. Weighted accuracy computes the accuracy based on sample weight for each class, which is more suitable for an imbalanced dataset. Precision and recall are commonly used in the evaluation metric for model performance. Precision represents the proportion of positive identifications that were actually correct. Recall indicates the proportion of actual positives that were identified correctly. F1 score is the harmonic mean of precision and recall, which is a measure that combines precision and recall. From Tables 1 and 2, we can see our dataset is imbalanced in that compared to normal cases; our dataset has fewer infected cases.

*10-fold cross-validation details.* We split data into 10 non-overlapping subsets; each time, we use one subset as a testing set and use the rest data as a training set. We set a 5 threshold for the infected prevalence based on a cross-validation experiment. For specific instances, it contains information: location, infected prevalence, etc. If the number of infected prevalence of a specific instance exceeds the threshold, we define it as Class 0 (which represents the infected class); otherwise, we define it as Class 1 (which represents the normal class) (Supplementary Figs. S3–S23). We repeat this process 10 times by taking a different training set and test set.

After the data is shuffled and split into training and testing sets, the experiments were carried out 10 times, the mean accuracy and the standard deviation were calculated, and training accuracy and testing accuracy for predicting dependent variables by different ML methods based on risk factors were generated. We take the average result as the final result. Cross-validation can overcome the overfitting problem.

**Ethical approval.** This study has been approved by the ethical committee of UNIVERSIDAD DE SONORA, Mexico. Informed consent was waived by the ethical committee because the data analyzed was aggregated, de-identified and delinked, and therefore, obtaining informed consent was not applicable.

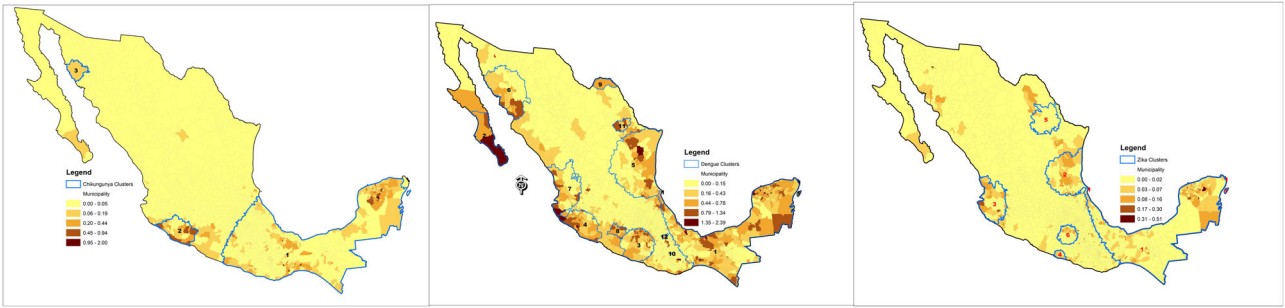

**Fig. 1 Spatial distribution of CHIKV, DENV, and ZIKV (2012–2019) in Mexico.** The blue lines indicate cluster boundaries, while the legend shades indicate the levels of prevalence. The darker the shade, the higher the prevalence. **A** Mean spatial distribution of Chikungunya. **B** Mean spatial distribution of Dengue. **C** Mean spatial distribution of Zika.

**Table 3 Standard error of classification performance of various ML algorithms on dengue, zika, and CHIKV virus prediction for all clusters.**

| Method | Dengue (standard error) | | | | | Zika (standard error) | | | | | CHIKV (standard error) | | | | |
|---|---|---|---|---|---|---|---|---|---|---|---|---|---|---|---|
| | Accuracy | Weighted accuracy | Precision | Recall | F-score | Accuracy | Weighted accuracy | Precision | Recall | F-score | Accuracy | Weighted accuracy | Precision | Recall | F-score |
| XGBoost | 0.08 | 0.07 | 0.05 | 0.06 | 0.07 | 0.04 | 0.13 | 0.16 | 0.18 | 0.07 | 0.04 | 0.19 | 0.15 | 0.19 | 0.16 |
| Decision tree | 0.08 | 0.09 | 0.11 | 0.09 | 0.11 | 0.01 | 0.11 | 0.09 | 0.11 | 0.14 | 0.07 | 0.12 | 0.11 | 0.18 | 0.21 |
| SVM | 0.07 | 0.13 | 0.05 | 0.10 | 0.15 | 0.05 | 0.13 | 0.18 | 0.15 | 0.11 | 0.08 | 0.09 | 0.25 | 0.23 | 0.13 |
| KNN | 0.04 | 0.18 | 0.05 | 0.13 | 0.22 | 0.07 | 0.15 | 0.21 | 0.28 | 0.17 | 0.09 | 0.18 | 0.23 | 0.14 | 0.19 |
| Random forest | 0.13 | 0.11 | 0.08 | 0.07 | 0.29 | 0.07 | 0.17 | 0.19 | 0.09 | 0.15 | 0.08 | 0.21 | 0.13 | 0.17 | 0.18 |
| Neural network | 0.13 | 0.09 | 0.11 | 0.13 | 0.19 | 0.07 | 0.08 | 0.15 | 0.19 | 0.13 | 0.17 | 0.23 | 0.11 | 0.27 | 0.23 |

## Results

**General disease patterns between 2012–2019.** DENV was the most prevalent of the three diseases throughout Mexico. Nearly 60.6% (1498/2469) of the municipalities reported DENV cases, 29.3% (723/2469) reported CHIKV cases, and 31.2% (771/2469) reported ZIKV cases. Of all the municipalities, 2.1% (52/2469) reported all three ABDs (Fig. 1). However, 39.6% (978/2469) of the municipalities in Mexico never reported any case of disease from these viral cases from 2012 through 2019. In total, 26,211 CHIKV, 224,701 DENV, and 12,813 laboratory-confirmed ZIKV cases were reported throughout the 8-year study period. In Mexico, 67 municipalities consistently reported more than 1% DENV prevalence, with the Tomatlán (Jalisco) municipality in the state of Jalisco reporting the highest prevalence (2.48%). A sharp increase in CHIKV, DENV, and ZIKV cases was reported in Veracruz.

Our results show that for all three ABDs, the influence of socio-economic attributes is larger than the influence of climate attributes for some clusters. This study shows that socio-economic features have more impact on the prevalence of ABDs in most areas, whereas, in other clusters, climatic features have the greatest impact. DENV is the most prevalent of the three diseases throughout Mexico, with over 60% of the municipalities reporting DENV cases, while only 29.3% reported CHIKV cases, and 31.2% ZIKV cases. Barely 2% of municipalities report all three ABDs. However, 39.6% (978/2469) of the municipalities in Mexico never reported any case of disease from these viral cases from 2012 through 2019. We find the attributes of altitude and minimum rainfall volume have a marginal influence on the model output. Average rainfall and maximum rainfall are more important than minimum rainfall.

**Spatiotemporal clusters.** Identified spatial clusters of CHIKV, DENV, and ZIKV prevalence are shown in Fig. 1. Twenty-one statistically significant ($p = 0.0001$) clusters were observed in Mexico. We analyzed all clusters and non-clusters, as SES features

and climate features may have different levels of impact for all clusters and non-clusters. Supplementary Table S1 indicates the majority vote and average results for CHIKV, DENV, and ZIKV prevalence based on different clusters. There were 12 spatio-temporal clusters of DENV prevalence (Supplementary Tables S1–S14). Climatic features had more impact than SES features on model output in clusters 1, 4, 5, 6, 7, and 12 (Supplementary Figs. S6–S17). There were six spatiotemporal clusters of ZIKV prevalence (Supplementary Tables S1, S15–S21). Climatic features had more impact in clusters 1, 2, 3, and 5, whereas SES features had more impact than climatic features on model output in clusters 4 and 6 (Supplementary Figs. S18–S23). There were three spatiotemporal clusters of CHIKV prevalence (Supplementary Tables S1, S22–S25). Climatic features had more impact than SES features on model output in clusters 1, 2, and 3 (Supplementary Figs. S3–S5). All model output data used to generate tables and figures are available as Supplementary Data.

Table 1 displays the performance of various ML classification algorithms across all clusters after taking the average. Table 3 demonstrates the standard error of classification performance of various ML algorithms on CHIKV, DENV, and ZIKV prevalence prediction for all clusters. Table 2 shows the performance of various ML classification algorithms across non-clusters after taking the average. Table 4 indicates the standard error of classification performance of various ML algorithms on CHIKV, DENV, and ZIKV case predictions for non-clusters. The results show that XGBoost performed the best in terms of precision-measure for CHIKV, DENV, and ZIKV prevalence (Tables 1 and 3). Besides XGBoost, other methods are baseline methods. The $F$ scores of XGBoost for CHIKV, DENV, and ZIKV prevalence are larger than other baseline approaches in most cases, which suggests XGBoost has better performance than other approaches (Table 1). The values of accuracy are larger than weighted accuracy and precision values (Tables 1 and 3). For instance, in Table 1, the accuracy of XGBoost under DENV prevalence is 0.93, which is higher than the corresponding weighted accuracy of 0.78 and precision of 0.86. This may happen due to a class imbalance issue. More specifically,

**Table 4 Standard error of classification performance of various ML algorithms on dengue, zika, and CHIKV virus prediction for non-clusters.**

| Method | Dengue (standard error) | | | | | Zika (standard error) | | | | | CHIKV (standard error) | | | | |
|---|---|---|---|---|---|---|---|---|---|---|---|---|---|---|---|
| | Accuracy | Weighted accuracy | Precision | Recall | *F*-score | Accuracy | Weighted accuracy | Precision | Recall | *F*-score | Accuracy | Weighted accuracy | Precision | Recall | *F*-score |
| XGBoost | 0.07 | 0.09 | 0.11 | 0.07 | 0.11 | 0.05 | 0.03 | 0.18 | 0.07 | 0.11 | 0.02 | 0.08 | 0.25 | 0.13 | 0.09 |
| Decision tree | 0.04 | 0.15 | 0.14 | 0.18 | 0.04 | 0.08 | 0.11 | 0.17 | 0.08 | 0.05 | 0.04 | 0.17 | 0.22 | 0.29 | 0.11 |
| SVM | 0.08 | 0.13 | 0.13 | 0.05 | 0.19 | 0.01 | 0.11 | 0.13 | 0.19 | 0.08 | 0.06 | 0.13 | 0.18 | 0.18 | 0.14 |
| KNN | 0.11 | 0.09 | 0.25 | 0.23 | 0.18 | 0.22 | 0.21 | 0.07 | 0.19 | 0.23 | 0.18 | 0.12 | 0.18 | 0.27 | 0.26 |
| Random forest | 0.08 | 0.11 | 0.17 | 0.12 | 0.17 | 0.08 | 0.19 | 0.17 | 0.19 | 0.21 | 0.06 | 0.34 | 0.12 | 0.21 | 0.25 |
| Neural network | 0.06 | 0.17 | 0.09 | 0.19 | 0.13 | 0.04 | 0.23 | 0.15 | 0.09 | 0.32 | 0.09 | 0.11 | 0.23 | 0.26 | 0.09 |

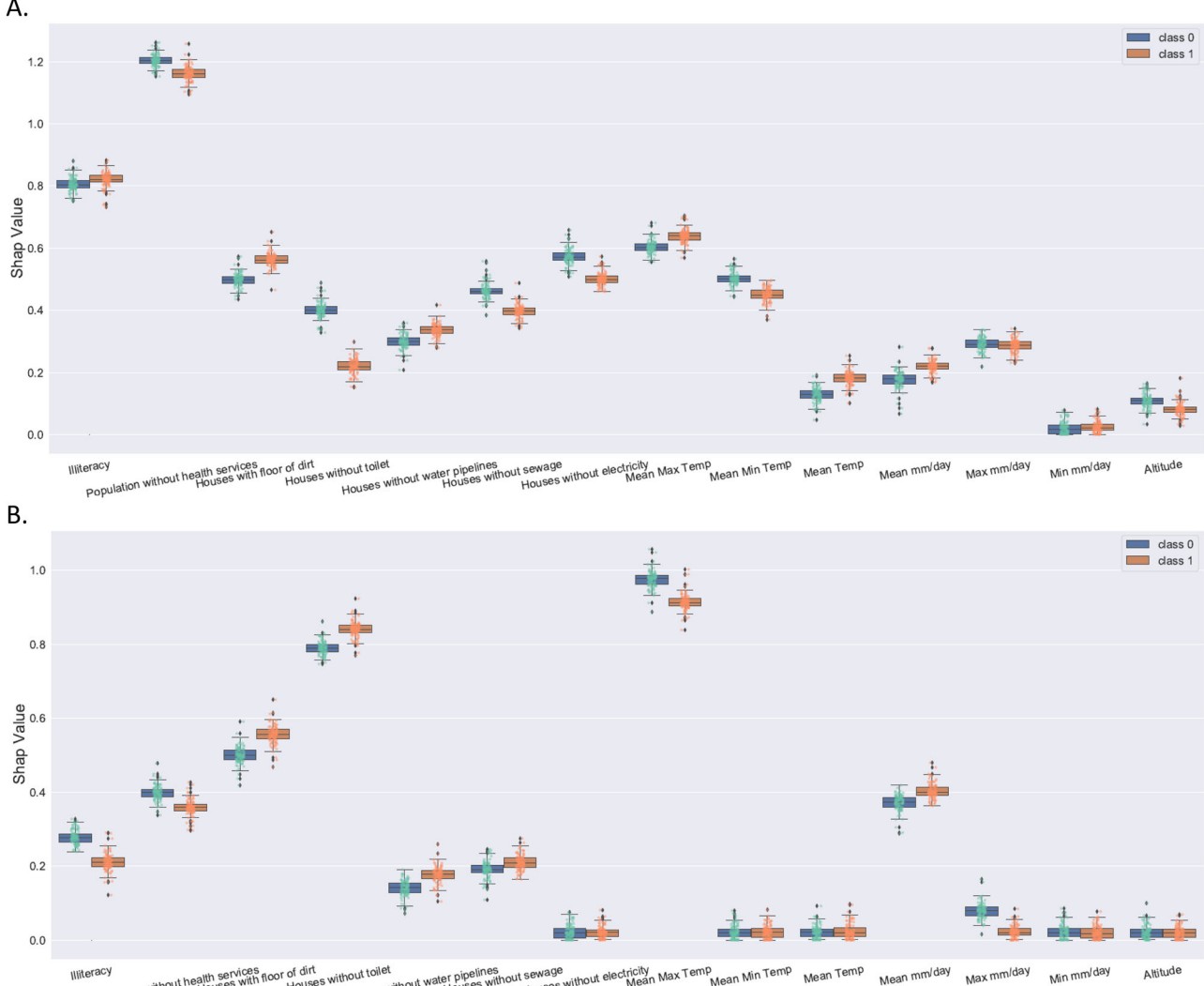

**Fig. 2 Average impact on model output magnitude for CHIKV using SHAP values.** Class 0 (blue) represents the infected class, while class 1 (brown) represents the normal class of CHIKV. **A** Average impact on model output magnitude for all clusters of CHIKV, and **B** Average impact on model output magnitude for all non-clusters of CHIKV.

concerning expected prevalence (normal class), we may have low prevalence.

Accuracy is the fraction of relevant correct instances over total instances. We presented the standard error of 'all clusters' and 'non-clusters' predictive results in accuracy, weighted accuracy, precision, recall (sensitivity), and *F* measure (Tables 2 and 4). XGBoost performed the best for CHIKV, DENV, and ZIKV prevalence. The standard error of XGBoost is more stable than other baseline approaches. For example, in Table 3, the standard error of XGBoost under DENV prevalence for

accuracy, weighted accuracy precision, recall, and *F*-score were 0.08, 0.07, 0.05, 0.06, and 0.07, respectively, which are lower than most of the standard errors of the other baseline approaches.

For all three ABDs, the influence of socio-economics attributes was larger than the influence of climate attributes for some clusters. Socio-economics attributes had a higher impact than climate attributes (Figs. 2A, B, 3A, B, 4A). The weighted socio-economic attributes SHAP value is 0.61, and the weighted climate attributes SHAP value is 0.39 (Fig. 4A).

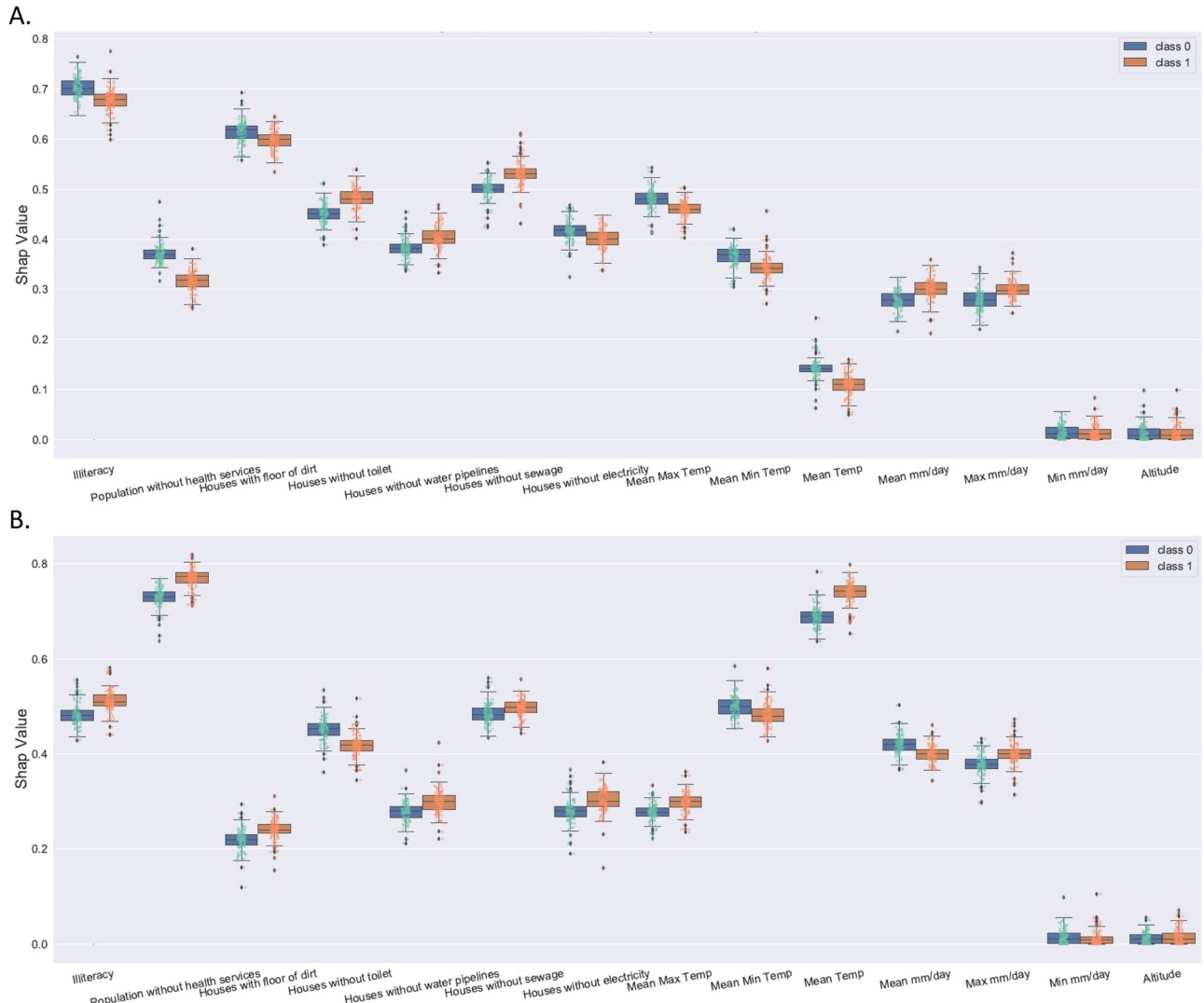

**Fig. 3 Average impact on model output magnitude for DENV using SHAP values.** Class 0 (blue) represents the infected class, while class 1 (brown) represents the normal class of DENV. **A** Average impact on model output magnitude for all clusters of DENV, and **B** Average impact on model output magnitude for all non-clusters of DENV.

The attributes of altitude and minimum rainfall volume had a marginal influence on the model output (Figs. 3A, B, 4A, B). The average rainfall volume and the maximum rainfall volume were more important than the minimum rainfall volume concerning the model output for the rainfall volume (Supplementary Fig. S1).

Based on the results presented in Tables 1–4, the accuracy of different approaches was higher than the corresponding weighted accuracy, precision, recall, and F-score. For example, in Table 1, for the decision tree method under the DENV case scenario, the accuracy was 0.91, while the corresponding weighted accuracy, precision, recall, and F-score were 0.78, 0.77, 0.78, and 0.78, respectively.

While the magnitude of measures of associations was slightly stronger for urban areas than for rural areas, the results show no differences in inferences. Regarding temperature, for example, the association with DENV outcome was relatively higher in urban than in rural areas. For population density, the association with DENV outcome was also slightly higher in urban than in rural areas. Inferences for urban in comparison with rural were similar for CHIKV and ZIKV.

## Discussion

This study set out to determine the longitudinal dynamics of three major arbovirus diseases over 8 years in Mexico. We found substantial differences in the prevalence of CHIKV, DENV, and ZIKV across Mexico. Tomatlán (Jalisco) had the highest level of DENV prevalence among all diseases. Acapulco de Juárez (Oaxaca) had the highest prevalence of CHIKV and ZIKV. Both climatic and SES attributes were significantly associated with risk factors of clustering of all three ABDs.

The outbreak of CHIKV and ZIKV in 2016 established a co-transmission of three different ABDs in certain municipalities in Mexico[47]. However, the circulation of all three viruses in the same municipalities at the same time continues to provide challenges and is concerning for public health[47]. The clinical presentations of CHIKV, DENV, and ZIKV are very similar, causing mis-identification when laboratory testing is not conducted[5]. This is important to keep in mind despite the prevalence analyzed in this study being laboratory-confirmed cases. The differences in each disease prevalence might be due to differences in landscape, vector control program, and socio-economic development for different locations in Mexico.

A.

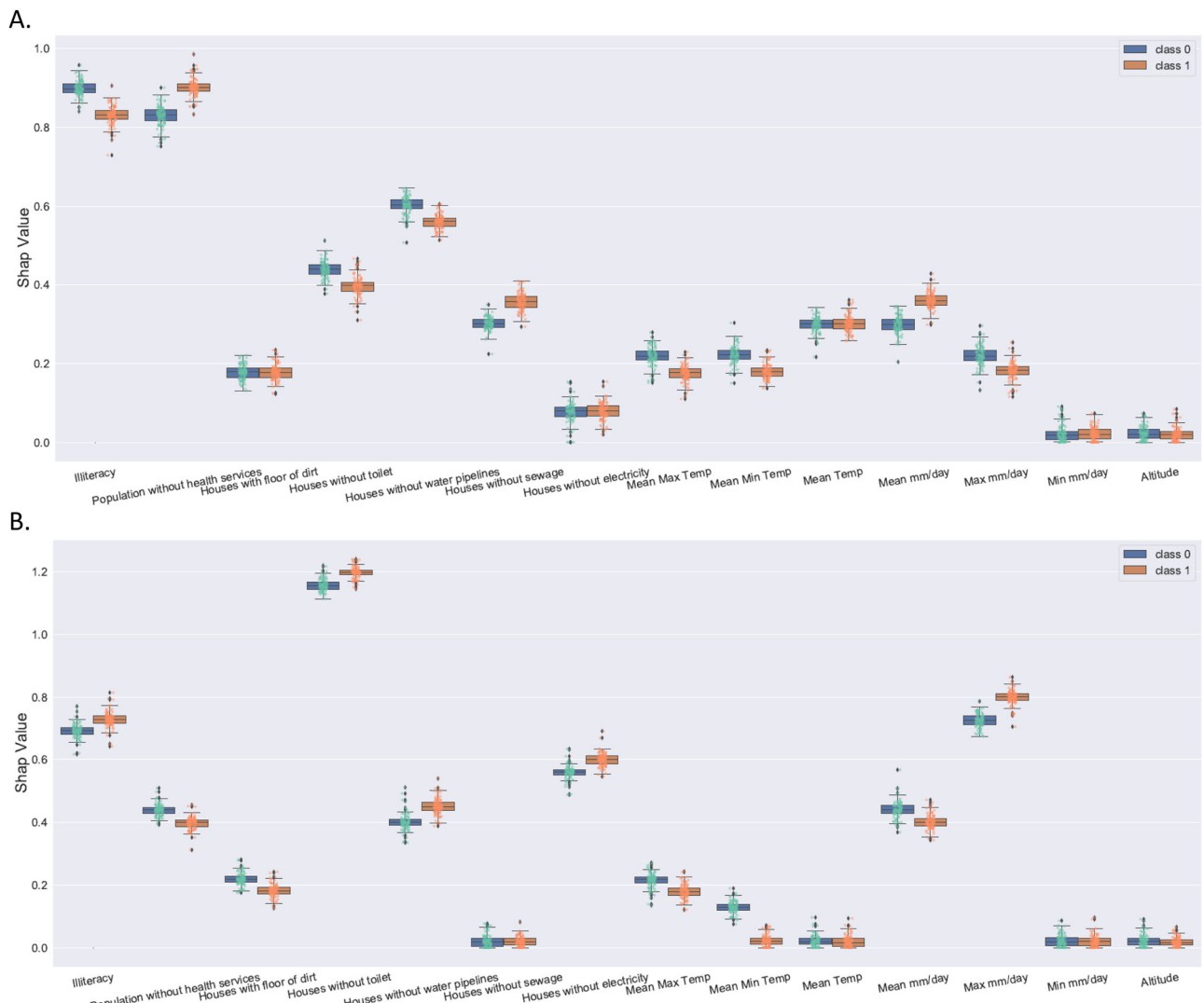

B.

**Fig. 4 Average impact on model output magnitude for ZIKV using SHAP values.** Class 0 (blue) represents the infected class, while class 1 (brown) represents the normal class of ZIKV. **A** Average impact on model output magnitude for all clusters of ZIKV, and **B** Average impact on model output magnitude for all non-clusters of ZIKV.

We found a positive association between mean temperature and CHIKV (Supplementary Fig. S6), DENV (Supplementary Figs. S7, S11, S12, S14–S16, S19), and ZIKV transmission (Supplementary Figs. S21 and S23), consistent with previous studies' findings[8,15,24,26,27,32]. While other published findings show that *Aedes* mosquitoes can be infected with and can transmit all combinations of these viruses simultaneously within the observed temperature ranges in Mexico[48–50], our results indirectly support this as evidence in the concurrent circulation of various arboviruses within the same population and geographic areas.

Clustering of CHIKV, DENV, and ZIKV prevalence has been associated with lower socioeconomic status (Supplementary Fig. S13), as indicated in the highest mean among houses without toilet facilities. Houses without toilet facilities, water pipelines, and access to improved sources of water are all conducive to creating an aquatic container habitat that harbors *Ae. aegypti* mosquito larvae. These socio-economic risk factors were highest (Supplementary Figs. S12, S13), where there were also high levels of illiteracy and consistent with the previous findings (Supplementary Figs. S13, S15, and S19).

In this work, we evaluated different ecosystems in 21 statistically significant clusters of three major arboviruses. CHIKV was

the only disease for which spread was clustered only in certain parts of Mexico. Twelve clusters had the greatest disease prevalence, having the most favorable climatic factors (Supplementary Figs. S1–S4, S7–S10, S15–S17, and S20), and 10 had greater poverty indices. These results highlight how climatic and socio-demographic factors are not uniformly predictive of ABDs throughout Mexico. The transmission of ABDs is complex and numerous factors could contribute to transmission heterogeneity. For example, the primary vector, *Ae. aegypti* is not uniformly distributed in Mexico and, in some cases, overlaps with *Ae. albopictus*, the secondary vector[4]. Additionally, a substantial proportion of Mexicans have naturally acquired antibodies from past exposure resulting in protective immunity for CHIKV, ZIKV, or the same DENV serotype[51].

A major contribution of this study is the implementation and comparison of spatial statistics and different machine learning techniques. The combination of these techniques helped to improve our understanding of the risk posed by these three viruses. The geographical settings for these clusters determined different climatic zones, different ecosystems, and variations in SES in Mexico. Specific climatic factors associated here highly affected disease prevalence. This was particularly evident for 11

out of 21 spatiotemporal clusters across Mexico. These different ecosystems are expected to establish contrasting socio-ecological and behavioral patterns that transmit ABDs. Nearly 39.6% of municipalities in Mexico never reported transmission of these three arboviruses. This makes sense as higher elevation regions of north-western central Mexico have fewer *Ae. aegypti* and lower reports of ABDs[52].

XGBoost performed the best in terms of precision (positive predictive value)-measure for these three ABDs. Compared to traditional machine learning methods (linear regression, logistic regression, naïve Bayes, *k*-means, decision trees, etc.), XGBoost used more accurate approximations to find the best tree model. XGBoost computed second-order gradients, i.e., second partial derivatives of the loss function, which provided more information about the direction of gradients and how to get to the minimum of our loss function. While regular gradient boosting used the loss function of our base model (e.g., decision tree) as a proxy for minimizing the overall model's error, XGBoost used the second-order derivative as an approximation. More details about the advantages and disadvantages of all machine learning methods are in the supplement section.

Findings from this study suggest that both climate and SES variables proved to be strong predictors in some clusters. However, for some clusters, either climate or SES variables proved to be a strong predictor.

The values of accuracy were higher than weighted accuracy and precision values, possibly due to the imbalanced scenario (the prevalence of people with infection is low and the data is sparse). The accuracy only considered the correct predicted cases and not which class the case came from. Therefore, if the dataset is imbalanced, we need to add more evaluation metrics to evaluate the model performance. In our results, we used weighted accuracy, precision, recall, and *F*-score to evaluate our models. Here we used aggregated data at the municipality level. This study also used only the laboratory-confirmed cases. Additionally, the risk factors for ABDs transmission were determined based on the passive surveillance system. In this study, we used municipality-level data and adjusted it for population density, seasonality, and presence of *Ae. aegypti*, *Ae. albopictus*, rural/urban classification, and altitude. Although the municipality-level data has been widely used[47], it could be possible that some of the observed patterns are confounded by potential hidden factors in our data, as the many factors at the individual and household level may influence the distribution of *Ae aegypti*. Identifying and addressing these hidden factors could be of great interest in future studies.

The distribution of ABDs infections is often driven by local spatiotemporal patterns influenced by fine-scale socio-economic, environmental, virological, and demographic factors[53–55]. The current analysis at the municipality scale is too crude to capture many of these drivers of transmission heterogeneity[56]. This implies a clear need for the development of a more integrated individual and household level with fine-scale time series data to understand the implications of these household patterns for targeted disease surveillance and vector control activities.

Future studies should be used to build predictive models to anticipate the time and location of ABDs outbreaks and determine the stand-alone influence of individual risk factors and establish causal relationship. Incorporating microclimate data, landscape ecology, and urban environment into disease transmission models has the potential to yield more spatial precision and ecologically interpretable metrics of mosquito-borne disease transmission risk in urban landscapes[57–59]. Further study of disease clusters concurrent with entomological data on *Aedes* distribution and human contact would also be beneficial. A better understanding of the drivers of ABDs transmission that consider local dynamics should contribute to the design of more effective mosquito control and disease prevention programs and promote public health in Mexico and other endemic countries.

**Reporting summary**. Further information on research design is available in the Nature Research Reporting Summary linked to this article.

## Data availability

The arbovirus data (Chikungunya, dengue, and Zika virus) used in this study are not publicly downloadable but can be requested at their original sites. Parties interested in data access should visit the Mexican Ministry of Health website (https://www.gob.mx/salud/en, E-mail: petitionscitizens@salud.gob.mx). The source data for the figures are available in Supplementary Data (Excel).

## Code availability

Code to reproduce study findings is freely available and accessible at GitHub link: https://github.com/BoDong111/COMMSMED in zenodo submission (https://doi.org/10.5281/zenodo.7071115)[60].

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

## Acknowledgements
U.H. was supported by the Research Council of Norway (grant #281077).

## Author contributions
Conceptualization: B.D., J.T., L.K., U.H. Methodology: B.D., L.K., B.Z., L.K., U.H. Investigation: B.D., L.K., U.H. Visualization: B.D. Supervision: L.K., U.H. Writing—original draft: B.D., L.K., U.H. Writing—review & editing: L.K., M.S., J.T., B.Z., G.L.H., U.A.L.L., A.A.M., J.L., U.S.D.T.N., U.H.

## Competing interests
The authors declare no competing interests.
