## [Peer Review File · Communications Medicine]

Reviewers' comments:

Reviewer #1 (Remarks to the Author):

This paper addresses the important question of identifying the role of social and climactic factors on the spread of three arboviruses. The paper proposes a variety of methods, which could help improve our understanding of these viruses and how they spread. However, as currently presented the study is incomplete, and does not provide a clear research objective or answer. Some specific concerns are listed below:

1. In general, there is very minimal discussion, interpretation and comparison of the findings, including how they compare against previous literature. What are the learnings from this study that can be useful for improving our understanding of the risk posed by these three viruses. How do they differ as a function of the considered risk factors? For instance, they authors state which variables have an impact for some diseases and some clusters, but there is no discussion of what type of impact and/or the mechanism that may be driving the impact.
2. A potential large contribution of this work is the implementation and comparison of so many methods. However, there is minimal discussion comparing the different methods presented, and the potential value each may add. What are the research questions and findings that each method might be well suited for? Which fail? Why might that be?
3. The clustering method and evaluation of the clusters identified is unclear, and needs to be better described and motivated, especially since the entire analysis is based on separate evaluation of clusters and non-clusters.
4. What is the reason for modeling the role of the variables for clusters and non-clusters separately?
5. It appears that SES and climate variables have different impacts across clusters and vs non-clusters. It's unclear whether these differences are actually significant. And why do these differences occur (e.g., why would mean temperature have a greater effect for nonclusters than for clusters?). The authors need to provide some discussion and possible underlying mechanisms for these findings, and well as confidence in them.
6. Further, why might the impact of these variables vary across the three diseases, which are spread by the same mosquito species?
7. It is unclear if the authors accounted for seasonality in their analysis. If not, this is likely impacting their findings on the role of climate variables.
8. The lit review is very light with respect to the findings, e.g. the authors say that dengue risk is "influenced by rainfall, temperature, relative humidity, and unplanned rapid urbanization" (line 99-100), but how do those variables have an impact? Same for the SES variables.
9. There needs to be more provided on the data sets and variables considered. E.g., what is the spatial distribution of climate zones and the socioeconomic variables? It would help to include a map or two of Mexico that provides this context?

The figures need further clarity:

10. Figure 1 seems to show the spatial clusters as circles and dengue cases as dots, but this isn't explicitly indicated in the caption or the text. I'm also having a hard time seeing a relationship between spatial cluster location/size and the distribution of the dengue cases. How are these identified and validated as clusters? Also, here it would help to plot dengue cases normalized to municipality population rather than showing the raw counts. Because as it is it looks like the dots are essentially mirroring a population map.

11. In Figure 2-4, what's the difference between Class 0 and Class 1? I couldn't find any mention of that in the text.

Reviewer #2 (Remarks to the Author):

This manuscript describes the findings from an extensive analysis of municipality-level case count data of dengue, chikungunya and Zika viruses reported to the national surveillance system and confirmed by laboratory analyses. The dataset is impressive, and the authors go in great detail analyzing it to: 1) identify spatio-temporal clusters of infection by each virus, and b) characterize potential environmental and socio-economic variables associated with the finding of such clusters. Despite the extensive application of classification algorithms to find such associations, the manuscript has many issues regarding the analysis of data and interpretation of findings.

General Comments

One of the major concerns with this article relates to a key assumption in spatial analysis. Spatial stationarity is an important (and very often untested) assumption in these types of analyses. When the mean of the process in question (as well as its variance) does vary throughout space (or space-time), one may find clusters due to this variance rather than a true process leading to the observed pattern. I am certain that the dataset analyzed has evidence of spatial stationarity. Aede-borne viruses are not transmitted in rural areas with low population density, or in high elevation areas, due to constraints on the viral replication and environmental suitability for *Aedes aegypti*. Therefore, the finding of clusters may just not be due to the occurrence of points but also to the marked difference in risk (and case counts) between rural and urban areas. The test assumes as a null hypothesis that case counts are proportional to population size. But, transmission is different from proportional as population size is reduced, given that many factors may influence *Ae aegypti* distribution, beyond population. If a specific process is driving the finding of municipalities to be members of clusters (or, conversely, never be identified as a cluster) then such process is responsible for the pattern. In this case, rurality and population density both would create a marked separation in risk and case counts. Due to this lack of stationarity, the finding of clusters is neither informative nor useful. This difference between settings in risk led to subsequent findings that are not surprising. Temperature would limit the vector, combined to the extent of urban areas. Other variables are clearly associated with rurality (dirt floors), and likely are just identifying the known fact that aedes-borne viruses are more urban than rural.

While this article focuses on statistical methods and validations, it lacks in biological understanding

and interpretation. There is no mention of the value of 'strength' between indices when reported (how biologically relevant is having a Xunit difference?). It is fine to know that a given set of variables is associated with case counts, but I believe it would be more important to report what ranges of values are associated with observed patterns. What are the temperature values associated with clusters versus non-clusters? Based on the figures 2-4 (which, by the way, report mean without showing the 95%CI) the difference between clusters and non-clusters for a given factor appears to be minor. Omitting such information turns this paper into a mere statistical exercise. I would not be surprised that when the values are presented, the differences are not only minor but also biased to the inclusion of rural areas (in cities, population density, temperature, may be less relevant than when including rural and even mountainous areas).

Other (less crucial, yet important) issues are listed below:

The article is heavily focused on findings and analyses by the group, and ignores key knowledge and references in the subject (both in the background and discussion). There is plenty of evidence associating temperature with arbovirus transmission. In fact, temperature variability (range between min and max daily temperature) helps explain spatial variability in incidence at a finer scale than temperature alone. There are good descriptions of dengue distribution in Mexico at the municipal level and of co-circulation and overlap of the three Aedes-borne viruses at the city level which are also not mentioned here.

The reporting of results is very focused on the classification algorithms than in the key finding of spatio-temporal clusters (which leads to the classification). What was the duration of each spatio-temporal cluster? Was there any overlap in municipalities being clusters for 2 or more viruses? Why not trying ellipses rather than circles? [I believe the south cluster that has more area outside than inside the circle is a good example of bias].

No limitations are mentioned in the discussion. For instance, relying on lab confirmed cases would bias the data towards areas that do more testing, have more population or better infrastructure. Not acknowledging this is an issue.

A figure with the incidence by municipality would have been very informative.

It is not clear whether the environmental and demographic conditions used for characterizing the separation between cluster and non-cluster areas involved the entire time series or only the period identified as spatio-temporal cluster. The latter would be much more appropriate, and it seems it was not what has been done.

Reviewer #3 (Remarks to the Author):

In COMMSMED-21-0143, Dong et al. analyzed spatio-temporal data related to Dengue, Chikungunya, and Zika viruses in Mexico.

S1. The authors made good use of SaTScan to detect spatial clusters.

S2. They also made good use of Pearson correlation coefficient and randomized dependence coefficient to analyze the socio-demographic and climatic impacts brought by these viruses and

diseases.

S3. They applied six existing machine learning models for clustering.

S4. They also used SHapley Additive exPlanations for interpretation of results.

S5. This showcases an application of machine learning in disease analytics.

W1. More details are needed for the Discussion section.

W2. What are the hypotheses?

W3. What are the recommendations based on the results from this case study?

COMMSMED-21-0143

Reviewer #1

This study does not provide a clear research objective or answer.

Response: The following aim statement is included on line 105-106: “This study aims to detect the spatiotemporal clusters and determine the risk factors of CHIKV, DENV, and ZIKV clusters in Mexico.”

In general, there is very minimal discussion, interpretation, and comparison of the findings, including how they compare against previous literature. What are the learnings from this study that can be useful for improving our understanding of the risk posed by these three viruses. How do they differ as a function of the considered risk factors? For instance, they authors state which variables have an impact for some diseases and some clusters, but there is no discussion of what type of impact and/or the mechanism that may be driving the impact.

Response: We have added the following paragraph to the discussion to compare the results of this study to previous literature:

“We found a positive significant association between mean temperature (Figures S4, S5, S6, S8, S9, S10, S11) and DENV, consistent with previous studies’ findings, (7, 14 - 16). We found a significant association between mean temperature (Figures S22, S23) with CHIKV, consistent with findings of previous studies (10, 20). We also found a significant association between mean temperature (Figure S18) with ZIKV transmission, consistent with findings of previous studies (10, 12). Findings from this study suggest that both climate and SES variables proved as a strong predictor in some clusters. However, for some clusters, either climate or SES variables proved as a strong predictor (lines: 331 -340)”.

A potential large contribution of this work is the implementation and comparison of so many methods. However, there is minimal discussion comparing the different methods presented, and the potential value each may add. What are the research questions and findings that each method might be well suited for? Which fail? Why might that be?

Response: We have added the following texts in the supplement file (lines 99 – 143).

KNN: Neighbours-based classification is a type of lazy learning as it does not attempt to construct a general internal model, but simply stores instances of the training data. Classification is computed from a simple majority vote of the k nearest neighbors of each point.

Advantages: This algorithm is simple to implement, robust to noisy training data, and effective if training data is large.

Disadvantages: Need to determine the value of K and the computation cost is high as it needs to compute the distance of each instance to all the training samples.

Decision Tree: Given data of attributes together with its classes, a decision tree produces a sequence of rules that can be used to classify the data.

Advantages: The decision Tree is simple to understand and visualize, requires little data preparation, and can handle both numerical and categorical data.

Disadvantages: Decision trees can create complex trees that do not generalize well, and decision trees can be unstable because small variations in the data might result in a completely different tree being generated.

Random Forest: Random forest classifier is a meta-estimator that fits a number of decision trees on various sub-samples of datasets and uses an average to improve the predictive accuracy of the model and controls over-fitting. The sub-sample size is always the same as the original input sample size but the samples are drawn with replacement.

Advantages: Reduction in over-fitting and random forest classifiers are more accurate than decision trees in most cases.

Disadvantages: Slow real-time prediction, difficult to implement, and complex algorithm.

SVM: Support vector machine is a representation of the training data as points in space separated into categories by a clear gap that is as wide as possible. New examples are then mapped into that same space and predicted to belong to a category based on which side of the gap they fall.

Advantages: Effective in high dimensional spaces and uses a subset of training points in the decision function so it is also memory efficient.

Disadvantages: The algorithm does not directly provide probability estimates, these are calculated using an expensive five-fold cross-validation.

Neural network: is based on a collection of connected units or nodes called artificial neurons, which loosely model the neurons in a biological brain.

Advantages: can simulate complex functions includes linear and non-linear functions.

Disadvantages: need a lot of time for training. Need a lot of time for tuning parameters.

XGBoost stands for Extreme Gradient Boosting; it is a specific implementation of the Gradient Boosting method which uses more accurate approximations to find the best tree model. It employs a number of nifty tricks that make it exceptionally successful, particularly with structured data. The most important are

1. computing second-order gradients, i.e. second partial derivatives of the loss function (similar to Newton's method), which provides more information about the direction of gradients and how to get to the minimum of our loss function. While regular gradient boosting uses the loss function of our base model (e.g. decision tree) as a proxy for minimizing the error of the overall model, XGBoost uses the 2nd order derivative as an approximation.

The clustering method and evaluation of the clusters identified is unclear, and needs to be better described and motivated, especially since the entire analysis is based on separate evaluation of clusters and non-clusters.

Response: We have added the following texts in the method section, supplement file (lines: 39-44).

“Under the null hypothesis, and in absence of covariates, it is expected that the number ABVs in each municipality is proportional to its population size. Poisson model requires the total counts of ABVs and population counts in each year and geographical coordinates for each municipality. The goal was to detect the statistically significant geographic clusters and identify the risk factors behind the clusters”.

What is the reason for modelling the role of the variables for clusters and non-clusters separately?

Response: This study aims to detect the spatiotemporal clusters and determine the risk factors of CHIKV, DENV, and ZIKV clusters in Mexico (lines: 105 - 106).

It appears that SES and climate variables have different impacts across clusters and non-clusters. It's unclear whether these differences are actually significant. And why do these differences occur (e.g., why would mean temperature have a greater effect for non-clusters than for clusters?). The authors need to provide some discussion and possible underlying mechanisms for these findings, and well as confidence in them.

Response: SES and climate variables have significantly different impacts across clusters and non-clusters. The reason might be *Ae. aegypti* and *Ae. albopictus* reported from certain parts of Mexico and overlap in some municipalities (1). However, the impact of the climatic and SES parameters vary across the three diseases might be due to the diversified climate zones and the wealth distribution in Mexico.

Further, why might the impact of these variables vary across the three diseases, which are spread by the same mosquito species?

Response: We have added the following texts:

“For example, the primary vector, *Ae. aegypti*, is not uniformly distributed in Mexico and in some cases, overlaps with *Ae. albopictus*, the secondary vector (4). Additionally, a substantial proportion of Mexicans have naturally-acquired antibodies from past exposure resulting in protective immunity for CHIK, ZIKV, or the same DENV serotype (49). Our current analysis utilizes human cases proportion to the entire population of a municipality and does not consider the proportion of the population that are not susceptible” (lines: 358-364).

It is unclear if the authors accounted for seasonality in their analysis. If not, this is likely impacting their findings on the role of climate variables.

Response: Besides climate variables, we also have socio-economic variables in our features. Even though climate features may have some seasonal properties, the socio-economic variables are marginally affected by seasonality. When we consider features, which combined with socio-economic features and climate features, it is hard to say the features have seasonality. We also converted the daily temperature and rainfall and aggregated at the monthly data which we believe is much finer than seasonal aggregation.

The lit review is very light with respect to the findings, e.g. the authors say that dengue risk is "influenced by rainfall, temperature, relative humidity, and unplanned rapid urbanization" (line 99-100), but how do those variables have an impact? Same for the SES variables.

Response: We have updated the lit review and added the following texts in the introduction (lines 69 – 79).

“Smaller to larger dengue outbreaks were associated with increased temperature (23.8°C to 33.1°C) and the delayed effects could be predicted with a one-week lag (2). Mean (>27°C), minimum (>22°C) and maximum temperature (>38°C) was found to be the most favorable weather condition at a lag of one to three months in the tropical and subtropical climate zone, respectively (3, 4).

Monthly mean rainfall showed a positive significant correlation with monthly dengue cases at a lag of 1–3 months (3-5). An increase of 1 cm of rainfall with a lag of two to three weeks was associated with dengue 1.3% to 2.1% of more dengue cases (6). Another study also suggested an increase of 1% of rainfall corresponded to an increase of 3.3% in the dengue cases (5).

Socio-economic factors such as barriers to healthcare services, inadequate sanitation, poverty, living in a poor neighborhood, and poor water supply were associated with the transmission of ABVs (7-10)”.

There needs to be more provided on the data sets and variables considered. E.g., what is the spatial distribution of climate zones and the socioeconomic variables? It would help to include a map or two of Mexico that provides this context?

Response: We have added two figures based on climate and SES data in Mexico (Figure S1 and Figure S2).

Figure 1 seems to show the spatial clusters as circles and dengue cases as dots, but this isn't explicitly indicated in the caption or the text. I'm also having a hard time seeing a relationship between spatial cluster location/size and the distribution of the dengue cases. How are these identified and validated as clusters? Also, here it would help to plot dengue cases normalized to municipality population rather than showing the raw counts. Because as it is it looks like the dots are essentially mirroring a population map.

Response: We have updated figure 1.

Here we used SaTScan (v. 9.6.1) to detect spatial clusters (settings: spatial analysis; discrete Poisson probability model; latitude/longitude coordinates; no geographical overlap; scanning for clusters with high rates). We assume the virus cases follow the discrete Poisson distribution. For each data instance, it contains information: location, infected cases, longitude, latitude, etc. Spatial clusters were determined by calculating the maximum likelihood ratio. Standardized prevalence ratios were estimated by dividing the number of observed cases by the number of expected cases in each cluster. Simulated p values were obtained using Monte Carlo methods.

In Figure 2-4, what's the difference between Class 0 and Class 1? I couldn't find any mention of that in the text.

Response: In our experiment, we set a threshold for the infected cases. For specific instances, it contains information: location, infected cases, etc. If the number of infected cases of a specific instance exceeds the threshold, we defined it as Class 0 (which represented the infected class), otherwise, we defined it as Class 1 (which represents the normal class) (lines: 286 – 289).

Reviewer #2

I am certain that the dataset analyzed has evidence of spatial stationarity. Aedes-borne viruses are not transmitted in rural areas with low population density, or in high elevation areas, due to constraints on the viral replication and environmental suitability for Aedes aegypti. Therefore, the finding of clusters may just not due to the occurrence of points but also to the marked difference in risk (and case counts) between rural and urban areas. The test assumes as a null hypothesis that case counts are proportional to population size. But, transmission is different from proportional as population size is reduced, given that many factors may influence Ae aegypti distribution, beyond population. If a specific process is driving the finding of municipalities to be members of clusters (or, conversely, never be identified as a cluster) then such process is responsible for the patten. In this case, rurality and population density both would create a marked separation in risk and case counts. Due to this lack of stationarity, the finding of clusters is neither informative nor useful.

Response: We thank the reviewer for raising this very important point. However, we apologized to the reviewer since we do not have any rural data. We used aggregated data at the municipality level and the mean center of each municipality was used in cluster detection. There was no selection bias (neither urban nor rural).

We have added the following texts in the limitations (lines: 381 – 386).

In this study, we also used municipality-level data and adjusted it for population density. Although the municipality level data has been widely used (11), it could be possible that some of the observed patterns are confounded by potential hidden factors in our data, as the many factors

(e.g., rurality) may influence the distribution of *Ae aegypti*. Identifying and addressing these hidden factors could be of great interest in future studies.

This difference between settings in risk led to subsequent findings that are not surprising. Temperature would limit the vector, combined to the extent of urban areas. Other variables are clearly associated with rurality (dirt floors), and likely are just identifying the known fact that aedes-borne viruses are more urban than rural.

Response: In this study, we used eight years' time series data. We hypothesized there are geographic clusters of CHIKV, DENV, and ZIKV in Mexico. For some clusters, socio-economic features may have more impact on the incidences of ABVs, whereas in other clusters are climatic features that have the greatest impact. To the best of our knowledge, no such location analysis has ever been published in Mexico (lines: 99 – 102).

While this article focuses on statistical methods and validations, it lacks in biological understanding and interpretation. There is no mention of the value of 'strength' between indices when reported (how biologically relevant is having a unit difference?). It is fine to know that a given set of variables is associated with case counts, but I believe it would be more important to report what ranges of values are associated with observed patterns. What are the temperature values associated with clusters versus non-clusters? Based on the figures 2-4 (which, by the way, report mean without showing the 95%CI) the difference between clusters and non-clusters for a given factor appears to be minor. Omitting such information turns this paper into a mere statistical exercise. I would not be surprised that when the values are presented, the differences are not only minor but also biased to the inclusion of rural areas (in cities, population density, temperature, may be less relevant than when including rural and even mountainous areas).

Response: The temperature features for cluster and non-cluster are same. Those are temperature related features in dataset.

In this study, we used aggregated data at the municipality level and the mean centre of each municipality was used in cluster detection. There was no selection bias.

We have added the following texts:

“Our results are also consistent with other published findings that these mosquitoes can be infected with and can transmit all combinations of these viruses simultaneously within the observed temperature ranges in Mexico (12-14)” (Lines: 338 – 340).

The article is heavily focused on findings and analyses by the group, and ignores key knowledge and references in the subject (both in the background and discussion). There is plenty of evidence associating temperature with arbovirus transmission. In fact, temperature variability (range between min and max daily temperature) helps explain spatial variability in

incidence at a finer scale than temperature alone. There are good descriptions of dengue distribution in Mexico at the municipal level and of co-circulation and overlap of the three Aedes-borne viruses at the city level which are also not mentioned here.

Response: We have updated the lit review and added the following texts in the introduction (lines: 69 – 79).

Smaller to larger dengue outbreaks were associated with increased temperature (23.8°C to 33.1°C) and the delayed effects could be predicted with a one-week lag (2). Mean (>27°C), minimum (>22°C) and maximum temperature (>38°C) was found to be the most favorable weather condition at a lag of one to three months in the tropical and subtropical climate zone, respectively (3, 4).

Here we have added new references

Dzul-Manzanilla F, Correa-Morales F, Che-Mendoza A, Palacio-Vargas J, Sanchez-Tejeda G, Gonzalez-Roldan JF, Lopez-Gatell H, Flores-Suarez AE, Gomez-Dantes H, Coelho GE, da Silva Bezerra HS, Pavia-Ruz N, Lenhart A, Manrique-Saide P, Vazquez-Prokopec GM, 2021. Identifying urban hotspots of dengue, chikungunya, and Zika transmission in Mexico to support risk stratification efforts: a spatial analysis. *Lancet Planet Health* 5: e277-e285.

Santos JPC, Honorio NA, Barcellos C, Nobre AA. A Perspective on Inhabited Urban Space: Land Use and Occupation, Heat Islands, and Precarious Urbanization as Determinants of Territorial Receptivity to Dengue in the City of Rio De Janeiro. *Int J Environ Res Public Health*. 2020;17(18).

The reporting of results is very focused on the classification algorithms than in the key finding of spatio-temporal clusters (which leads to the classification). What was the duration of each spatio-temporal cluster? Was there any overlap in municipalities being clusters for 2 or more viruses? Why not trying ellipses rather than circles? [I believe the south cluster that has more area outside than inside the circle is a good example of bias].

Response: The duration of all significant clusters is for one specific year. We put an enormous amount of time to run this clustering algorithm and used the entire study period to detect statistically significant spatiotemporal clusters and select the best and most meaningful findings for the classification algorithms.

We have updated figure 1. Yes, there are certain locations where 2 or more viruses were reported.

There is no bias in area selection. The software detected only those municipalities that observed significant clusters. We used circles for visualization/ cover those municipalities rather than using multiple colors. Otherwise, the small municipalities on the map cannot be visualized and mislead.

No limitations are mentioned in the discussion. For instance, relying on lab-confirmed cases would bias the data towards areas that do more testing, have more population, or have better infrastructure. Not acknowledging this is an issue.

Response: We agree and have added limitations: “The values of accuracy were higher than weighted accuracy and precision values, possibly due to the imbalanced scenario. Compared to the normal cases, we had fewer cases of infection. The accuracy only considered the correct predicted cases and not which class the case come from. Therefore, if the dataset is imbalanced, we needed to add more evaluation metrics to evaluate the model performance. In our results, we used weighted accuracy, precision, recall, and F1 score to evaluate our models. Here we used aggregated data at the municipality level. This study also used only the laboratory-confirmed cases and potentially lost many positive cases. The risk factors for ABVs transmission were also determined based on the passive surveillance system” (lines: 374 – 381).

A figure with the incidence by municipality would have been very informative.

Response: We have added figure 1.

It is not clear whether the environmental and demographic conditions used for characterizing the separation between cluster and non-cluster areas involved the entire time series or only the period identified as Spatio-temporal cluster. The latter would be much more appropriate, and It seems it was not what has been done.

Response: To detect the geographical clusters, we used total counts of ABVs and population counts in each year, and geographical coordinates for each municipality. No environmental conditions were considered. Here only the period is identified as Spatio-temporal cluster.

Reviewer #3

More details are needed for the Discussion section.

Response: We have substantially expanded the discussion section (lines: 331-340, 358-364, 374-386, 396-404).

What are the hypotheses?

Response: We have added the following to lines 99-102:

“Thus in this study, we hypothesize there are geographic clusters of CHIKV, DENV, and ZIKV in Mexico. For some clusters, socio-economic features may have more impact on the incidences of ABVs, whereas in other clusters are climatic features that have the greatest impact.”

What are the recommendations based on the results from this case study?

Response: We have added the following text (lines: 396-404)

“The distribution of ABVs infections is often driven by local spatiotemporal patterns influenced by fine-scale socio-economic, environmental, virological, and demographic factors (15-17). This implies a clear need for the integration of fine-scale longitudinal data at the individual and household level to understand the implications of these patterns for targeted disease surveillance and vector control activities.

Future studies should be used to build predictive models to anticipate the time and location of ABVs outbreaks and determine the risk factors. This is useful in prioritizing vector-control programs, implementing and developing adaptation systems due to climate change. Further study of disease clusters along with *Aedes* mosquito distribution, should be undertaken. A better understanding of the drivers of ABV transmission that consider local dynamics, should contribute to the design of a more effective vector mosquito control program, disease prevention, and promote public health in Mexico and other endemic countries. A more detailed study about the impact of heatwave and dengue outbreaks at a microgeographic scale may reveal further insights and help to control outbreaks (2).

References

1. Lubinda J, Trevino CJ, Walsh MR, Moore AJ, Hanafi-Bojd AA, Akgun S, et al. Environmental suitability for *Aedes aegypti* and *Aedes albopictus* and the spatial distribution of major arboviral infections in Mexico. *Parasite Epidemiol Control*. 2019;6:e00116.
2. Cheng J, Bambrick H, Yakob L, Devine G, Frentiu FD, Toan DTT, et al. Heatwaves and dengue outbreaks in Hanoi, Vietnam: New evidence on early warning. *PLoS Negl Trop Dis*. 2020;14(1):e0007997.
3. Akter R, Hu W, Gatton M, Bambrick H, Naish S, Tong S. Different responses of dengue to weather variability across climate zones in Queensland, Australia. *Environ Res*. 2020;184:109222.
4. Bal, S., Sodoudi, S. Modeling and prediction of dengue occurrences in Kolkata, India, based on climate factors. *Int J Biometeorol* 64, 1379–1391 (2020).
5. Polwiang S. The time series seasonal patterns of dengue fever and associated weather variables in Bangkok (2003-2017). *BMC Infect Dis*. 2020;20(1):208.
6. Hurtado-Diaz M, Riojas-Rodriguez H, Rothenberg SJ, Gomez-Dantes H, Cifuentes E. Short communication: impact of climate variability on the incidence of dengue in Mexico. *Trop Med Int Health*. 2007;12(11):1327-37.
7. Morgan J, Strode C, Salcedo-Sora JE. Climatic and socio-economic factors supporting the co-circulation of dengue, Zika and chikungunya in three different ecosystems in Colombia. *PLoS Negl Trop Dis*. 2021;15(3):e0009259.
8. Rodrigues NCP, Daumas RP, de Almeida AS, Dos Santos RS, Koster I, Rodrigues PP, et al. Risk factors for arbovirus infections in a low-income community of Rio de Janeiro, Brazil, 2015-2016. *PLoS One*. 2018;13(6):e0198357.

9. Whiteman A, Loaiza JR, Yee DA, Poh KC, Watkins AS, Lucas KJ, et al. Do socioeconomic factors drive Aedes mosquito vectors and their arboviral diseases? A systematic review of dengue, chikungunya, yellow fever, and Zika Virus. *One Health*. 2020;11:100188.
10. Charette M, Berrang-Ford L, Coomes O, Llanos-Cuentas EA, Carcamo C, Kulkarni M, et al. Dengue Incidence and Sociodemographic Conditions in Pucallpa, Peruvian Amazon: What Role for Modification of the Dengue-Temperature Relationship? *Am J Trop Med Hyg*. 2020;102(1):180-90.
11. Dzul-Manzanilla F, Correa-Morales F, Che-Mendoza A, Palacio-Vargas J, Sanchez-Tejeda G, Gonzalez-Roldan JF, et al. Identifying urban hotspots of dengue, chikungunya, and Zika transmission in Mexico to support risk stratification efforts: a spatial analysis. *Lancet Planet Health*. 2021;5(5):e277-e85.
12. Kinney RM, Huang CY, Whiteman MC, Bowen RA, Langevin SA, Miller BR, et al. Avian virulence and thermostable replication of the North American strain of West Nile virus. *J Gen Virol*. 2006;87(Pt 12):3611-22.
13. Ruckert C, Weger-Lucarelli J, Garcia-Luna SM, Young MC, Byas AD, Murrieta RA, et al. Impact of simultaneous exposure to arboviruses on infection and transmission by Aedes aegypti mosquitoes. *Nat Commun*. 2017;8:15412.
14. Bellone R, Failloux AB. The Role of Temperature in Shaping Mosquito-Borne Viruses Transmission. *Front Microbiol*. 2020;11:584846.
15. Salje H, Lessler J, Maljkovic Berry I, Melendrez MC, Endy T, Kalayanarooj S, et al. Dengue diversity across spatial and temporal scales: Local structure and the effect of host population size. *Science*. 2017;355(6331):1302-6.
16. Stoddard ST, Forshey BM, Morrison AC, Paz-Soldan VA, Vazquez-Prokopec GM, Astete H, et al. House-to-house human movement drives dengue virus transmission. *Proc Natl Acad Sci U S A*. 2013;110(3):994-9.
17. Bonifay T, Douine M, Bonnefoy C, Hurpeau B, Nacher M, Djossou F, et al. Poverty and Arbovirus Outbreaks: When Chikungunya Virus Hits More Precarious Populations Than Dengue Virus in French Guiana. *Open Forum Infect Di*. 2017;4(4).

COMMSMED-21-0143

Reviewer #1 (Remarks to the Author):

This paper addresses the important question of identifying the role of social and climactic factors on the spread of three arboviruses. The paper proposes a variety of methods, which could help improve our understanding of these viruses and how they spread. However, as currently presented the study is incomplete, and does not provide a clear research objective or answer. Some specific concerns are listed below:

Response: Thank you very much. We have responded point by point below.

The following aim statement is included on line 38-40, and 116 - 117: “The objective of this study is to detect the spatiotemporal clusters and determine the risk factors of three major Aedes-borne diseases (chikungunya, dengue, and Zika) clusters in Mexico.”, “this study aims to detect the spatiotemporal clusters and determine the risk factors of CHIKV, DENV, and ZIKV clusters in Mexico”.

In general, there is very minimal discussion, interpretation, and comparison of the findings, including how they compare against previous literature. What are the learnings from this study that can be useful for improving our understanding of the risk posed by these three viruses. How do they differ as a function of the considered risk factors? For instance, they authors state which variables have an impact for some diseases and some clusters, but there is no discussion of what type of impact and/or the mechanism that may be driving the impact.

Response: We have added the following paragraph to the discussion to compare the results of this study to previous literature:

“We found a positive association between mean temperature and CHIKV (Figures S6), DENV (Figures S7, S11, S12, S14, S15, S16, S19), and ZIKV transmission (Figure S21, S23), consistent with previous studies’ findings (8, 15, 24, 26, 27, 32). Our results are also consistent with other published findings that these mosquitoes can be infected with and can transmit all combinations of these viruses simultaneously within the observed temperature ranges in Mexico (49-51) (lines: 376 -381)”.

A major contribution of this study is the implementation and comparison of spatial statistics and different machine learning techniques. The combination of these techniques helped to improve our understanding of the risk posed by these three viruses. The geographical settings for these clusters determined different climatic zones, different ecosystems, and variations in SES in Mexico. Specific climatic factors associated here significantly affected disease prevalence. This was particularly evident for 11 out of 21 spatiotemporal clusters across Mexico. These different ecosystems are expected to establish contrasting socio-ecological and behavioral patterns that transmit ABDs. Nearly 39.6% of municipalities in Mexico never reported transmission of these three arboviruses. This makes sense as higher elevation regions of north-western central in Mexico have fewer *Ae. aegypti* (4) and lower reports of ABDs (48) (lines: 404 -413)”.

XGBoost performed the best in terms of precision (positive predictive value)-measure for these three ABDs. Compared to traditional machine learning methods (linear regression, logistic regression, naïve Bayes, k-means, decision trees, etc.), XGBoost used more accurate approximations to find the best tree model. XGBoost computed second-order gradients, i.e., second partial derivatives of the loss function, which provided more information about the direction of gradients and how to get to the minimum of our loss function. While regular gradient boosting used the loss function of our base model (e.g. decision tree) as a proxy for minimizing the overall model's error, XGBoost used the second-order derivative as an approximation. More details about the advantages and disadvantages of all machine learning methods are in the supplement section (lines: 415 -424)".

Findings from this study suggest that both climate and SES variables proved as a strong predictor in some clusters. However, for some clusters, either climate or SES variables proved as a strong predictor (lines: 426 - 428)".

A potential large contribution of this work is the implementation and comparison of so many methods. However, there is minimal discussion comparing the different methods presented, and the potential value each may add. What are the research questions and findings that each method might be well suited for? Which fail? Why might that be?

Response: We have added the following paragraph to the discussion to compare the results of this study to previous literature:

A major contribution of this study is the implementation and comparison of spatial statistics and different machine learning techniques. The combination of these techniques helped to improve our understanding of the risk posed by these three viruses. The geographical settings for these clusters determined different climatic zones, different ecosystems, and variations in SES in Mexico. Specific climatic factors associated here significantly affected disease prevalence. This was particularly evident for 11 out of 21 spatiotemporal clusters across Mexico. These different ecosystems are expected to establish contrasting socio-ecological and behavioral patterns that transmit ABDs. Nearly 39.6% of municipalities in Mexico never reported transmission of these three arboviruses. This makes sense as higher elevation regions of north-western central in Mexico have fewer *Ae. aegypti* (4) and lower reports of ABDs (48) (lines: 404 – 413).

XGBoost performed the best in terms of precision (positive predictive value)-measure for these three ABDs. Compared to traditional machine learning methods (linear regression, logistic regression, naïve Bayes, k-means, decision trees, etc.), XGBoost used more accurate approximations to find the best tree model. XGBoost computed second-order gradients, i.e., second partial derivatives of the loss function, which provided more information about the direction of gradients and how to get to the minimum of our loss function. While regular gradient boosting used the loss function of our base model (e.g. decision tree) as a proxy for minimizing the overall model's error, XGBoost used the second-order derivative as an approximation. More details about the advantages and disadvantages of all machine learning methods are in the supplement section (lines: 415 - 424)".

We have also added the following texts in the supplement file (lines 99 – 143).

KNN: Neighbours-based classification is a type of lazy learning as it does not attempt to construct a general internal model, but simply stores instances of the training data. Classification is computed from a simple majority vote of the k nearest neighbors of each point.

Advantages: This algorithm is simple to implement, robust to noisy training data, and effective if training data is large.

Disadvantages: Need to determine the value of K and the computation cost is high as it needs to compute the distance of each instance to all the training samples.

Decision Tree: Given data of attributes together with its classes, a decision tree produces a sequence of rules that can be used to classify the data.

Advantages: The decision Tree is simple to understand and visualize, requires little data preparation, and can handle both numerical and categorical data.

Disadvantages: Decision trees can create complex trees that do not generalize well, and decision trees can be unstable because small variations in the data might result in a completely different tree being generated.

Random Forest: Random forest classifier is a meta-estimator that fits a number of decision trees on various sub-samples of datasets and uses an average to improve the predictive accuracy of the model and controls over-fitting. The sub-sample size is always the same as the original input sample size but the samples are drawn with replacement.

Advantages: Reduction in over-fitting and random forest classifiers are more accurate than decision trees in most cases.

Disadvantages: Slow real-time prediction, difficult to implement, and complex algorithm.

SVM: Support vector machine is a representation of the training data as points in space separated into categories by a clear gap that is as wide as possible. New examples are then mapped into that same space and predicted to belong to a category based on which side of the gap they fall.

Advantages: Effective in high dimensional spaces and uses a subset of training points in the decision function so it is also memory efficient.

Disadvantages: The algorithm does not directly provide probability estimates, these are calculated using an expensive five-fold cross-validation.

Neural network: is based on a collection of connected units or nodes called artificial neurons, which loosely model the neurons in a biological brain.

Advantages: can simulate complex functions includes linear and non-linear functions.

Disadvantages: need a lot of time for training. Need a lot of time for tuning parameters.

XGBoost stands for Extreme Gradient Boosting: it is a specific implementation of the Gradient Boosting method which uses more accurate approximations to find the best tree model. It employs a number of nifty tricks that make it exceptionally successful, particularly with structured data. The most important are

1. computing second-order gradients, i.e. second partial derivatives of the loss function (similar to Newton's method), which provides more information about the direction of gradients and how to get to the minimum of our loss function. While regular gradient boosting uses the loss

function of our base model (e.g. decision tree) as a proxy for minimizing the error of the overall model, XGBoost uses the 2nd order derivative as an approximation.

The clustering method and evaluation of the clusters identified is unclear, and needs to be better described and motivated, especially since the entire analysis is based on separate evaluation of clusters and non-clusters.

Response: We have added the following texts in the method section, and supplement section.

SaTScan (v. 9.6.1) was used to detect spatial clusters (adjusted for more likely clusters) separately for CHIKV, DENV, and ZIKV (settings: spatial analysis; discrete Poisson probability model; latitude/longitude coordinates; no geographical overlap; scanning for clusters with high rates). Spatial clusters were determined by calculating the maximum likelihood ratio. Standardized prevalence ratios were estimated by dividing the number of observed cases by the number of expected cases in each cluster. Simulated p-values were obtained using Monte Carlo methods with 9,999 replications (43) (lines: 187 – 193).

“Under the null hypothesis, and in absence of covariates, it is expected that the number ABDs in each municipality is proportional to its population size. Poisson model requires the total counts of ABDs and population counts in each year and geographical coordinates for each municipality. The goal was to detect the statistically significant geographic clusters and identify the risk factors behind the clusters” file (lines: 39-44, supplement).

What is the reason for modelling the role of the variables for clusters and non-clusters separately?

Response: This study aims to detect the spatiotemporal clusters and determine the risk factors of CHIKV, DENV, and ZIKV clusters in Mexico (lines: 116 - 117).

It appears that SES and climate variables have different impacts across clusters and vs non-clusters. It's unclear whether these differences are actually significant. And why do these differences occur (e.g., why would mean temperature have a greater effect for nonclusters than for clusters?). The authors need to provide some discussion and possible underlying mechanisms for these findings, and well as confidence in them.

Response: SES and climate variables have significantly different impacts across clusters and non-clusters. The reason might be *Ae. aegypti* and *Ae. albopictus* reported from certain parts of Mexico and overlap in some municipalities (1). However, the impact of the climatic and SES parameters vary across the three diseases might be due to the diversified climate zones and the wealth distribution in Mexico. In our revised analysis, we have added the presence of *Ae. aegypti* and *Ae. albopictus* in each municipality. We have also added two new figures (climate zone and socio economics maps) and additional texts (Findings from this study suggest that both climate and SES variables proved as a strong predictor in some clusters. However, for some clusters, either climate or SES variables proved as a strong predictor.) between lines 426 - 428.

Further, why might the impact of these variables vary across the three diseases, which are spread by the same mosquito species?

Response: We have added the following texts:

“For example, the primary vector, *Ae. aegypti*, is not uniformly distributed in Mexico and in some cases, overlaps with *Ae. albopictus*, the secondary vector (4). Additionally, a substantial proportion of Mexicans have naturally-acquired antibodies from past exposure resulting in protective immunity for CHIKV, ZIKV, or the same DENV serotype (49). Our current analysis utilizes human cases proportion to the entire population of a municipality and does not consider the proportion of the population that are not susceptible” (lines: 396 - 402).

It is unclear if the authors accounted for seasonality in their analysis. If not, this is likely impacting their findings on the role of climate variables.

Response: We have accounted for seasonality in analysis and added a figure S24 to indicate the impact for seasonality. In this figure, we plot the average confirmed cases for CHIKV, ZIKV and DENV from Jan to Dec over 2012-2019. From the figure we can see, seasonality has more impact on DENV. For DENV, the confirmed cases number increase in July to October. For CHIKV and ZIKV, the seasonality has marginally impact.

We have added the following texts

“We separately used CHIKV, DENV, ZIKV cases as an outcome variable in our dataset, and socio-economic variables, population density, urban/rural information, altitude, seasonality, and presence of *Ae. aegypti* and *Ae. albopictus* and climate variables as our features (predictor) to build the model” (lines: 202 – 205).

“In this study, we also used municipality-level data and adjusted it for population density, seasonality, presence of *Ae. aegypti*, *Ae. albopictus*, rural/ urban classification, and altitude” (lines: 437 – 439).

The lit review is very light with respect to the findings, e.g. the authors say that dengue risk is "influenced by rainfall, temperature, relative humidity, and unplanned rapid urbanization" (line 99-100), but how do those variables have an impact? Same for the SES variables.

Response: We have updated the lit review and added the following texts in the introduction (lines 88 – 96).

“Smaller to larger dengue outbreaks were associated with increased temperature (23.8°C to 33.1°C) and the delayed effects could be predicted with a one-week lag (18). Mean (>27°C), minimum (>22°C) and maximum temperature (>38°C) was found to be the most favorable weather condition at a lag of one to three months in the tropical and subtropical climate zone, respectively (19, 20). Monthly mean rainfall showed a positive correlation with monthly dengue cases at a lag of one to three months (19-21). An increase of one cm of rainfall with a lag of two

to three weeks was associated with 1.3 to 2.1% more dengue cases (22). Another study also suggested an increase of 1% in rainfall corresponded to an increase of 3.3% in the dengue cases (21).

In addition, socio-economic factors such as barriers to healthcare services, inadequate sanitation, poverty, living in a poor neighborhood, and poor water supply were associated with the transmission of ABDs (11-15) ” (lines 83 – 85).

There needs to be more provided on the data sets and variables considered. E.g., what is the spatial distribution of climate zones and the socioeconomic variables? It would help to include a map or two of Mexico that provides this context?

Response: We have added two figures based on climate and SES data in Mexico (Figure S1 and Figure S2).

Figure 1 seems to show the spatial clusters as circles and dengue cases as dots, but this isn't explicitly indicated in the caption or the text. I'm also having a hard time seeing a relationship between spatial cluster location/size and the distribution of the dengue cases. How are these identified and validated as clusters? Also, here it would help to plot dengue cases normalized to municipality population rather than showing the raw counts. Because as it is it looks like the dots are essentially mirroring a population map.

Response: We have updated figure 1.

Here we used SaTScan (v. 9.6.1) to detect spatial clusters (settings: spatial analysis; discrete Poisson probability model; latitude/longitude coordinates; no geographical overlap; scanning for clusters with high rates). We assume the virus cases follow the discrete Poisson distribution. For each data instance, it contains information: location, infected cases, longitude, latitude, etc. Spatial clusters were determined by calculating the maximum likelihood ratio. Standardized prevalence ratios were estimated by dividing the number of observed cases by the number of expected cases in each cluster. Simulated p values were obtained using Monte Carlo methods. (lines: 187 – 193).

In Figure 2-4, what's the difference between Class 0 and Class 1? I couldn't find any mention of that in the text.

Response: We set a threshold for the infected prevalence. For specific instances, it contains information: location, infected prevalence, etc. If the number of infected prevalence of a specific instance exceeds the threshold, we defined it as Class 0 (which represented the infected class), otherwise, we defined it as Class 1 (which represents the normal class) (Figure S3-S23) (lines: 226 – 229).

Reviewer #2

This manuscript describes the findings from an extensive analysis of municipality-level case count data of dengue, chikungunya and Zika viruses reported to the national surveillance system and confirmed by laboratory analyses. The dataset is impressive, and the authors go in great detail analyzing it to: 1) identify spatio-temporal clusters of infection by each virus, and b) characterize potential environmental and socio-economic variables associated with the finding of such clusters. Despite the extensive application of classification algorithms to find such associations, the manuscript has many issues regarding the analysis of data and interpretation of findings.

General Comments

One of the major concerns with this article relates to a key assumption in spatial analysis. Spatial stationarity is an important (and very often untested) assumption in these types of analyses. When the mean of the process in question (as well as its variance) does vary throughout space (or space-time), one may find clusters due to this variance rather than a true process leading to the observed pattern.

Response: We performed the spatial analysis (cluster detection) using a well-known methodology. This method is well known in spatial statistics (<https://www.satscan.org/>). To detect the geographical clusters, we used total counts of ABDs and population counts in each year, and geographical coordinates for each municipality.

Here we used SaTScan (v. 9.6.1) to detect spatial clusters (settings: spatial analysis; discrete Poisson probability model; latitude/longitude coordinates; no geographical overlap; scanning for clusters with high rates). We assume the virus cases follow the discrete Poisson distribution. For each data instance, it contains information: location, infected cases, longitude, latitude, etc. Spatial clusters were determined by calculating the maximum likelihood ratio. Standardized prevalence ratios were estimated by dividing the number of observed cases by the number of expected cases in each cluster. Simulated p values were obtained using Monte Carlo methods. (lines: 187 – 193).

I am certain that the dataset analyzed has evidence of spatial stationarity. Aedes-borne viruses are not transmitted in rural areas with low population density, or in high elevation areas, due to constraints on the viral replication and environmental suitability for Aedes aegypti. Therefore, the finding of clusters may just not be due to the occurrence of points but also to the marked difference in risk (and case counts) between rural and urban areas. The test assumes as a null hypothesis that case counts are proportional to population size. But, transmission is different from proportional as population size is reduced, given that many factors may influence Ae aegypti distribution, beyond population. If a specific process is driving the finding of municipalities to be members of clusters (or, conversely, never be identified as a cluster) then such process is responsible for the pattern. In this case, rurality and population density both would create a marked separation in risk and case counts. Due to this lack of stationarity, the finding of clusters is neither informative nor useful.

Response: We thank the reviewer for raising this very important point. We added the population density feature (which can also reflect urban/rural information) in our dataset and update corresponding results and plots. We also added altitude, presence of *Ae. aegypti* and *Ae. albopictus* as features in our dataset.

We conducted a stratified analysis to address the potential biases and examined the magnitude and 95% CI in the associations of predictors (independent variables) with CHIKV, DENV, and ZIKV outcomes separately for urban and rural areas (lines: 231 – 233).

While the magnitude of measures of associations was slightly stronger for urban areas than for rural areas, the results show no differences in inferences. Regarding temperature, for example, the association with DENV outcome was relatively higher in urban than in rural areas. For population density, the association with DENV outcome was also slightly higher in urban than in rural areas. Inferences for urban in comparison with rural were similar for CHIKV and ZIKV (lines: 339 – 343).

This difference between settings in risk led to subsequent findings that are not surprising. Temperature would limit the vector, combined to the extent of urban areas. Other variables are clearly associated with rurality (dirt floors), and likely are just identifying the known fact that aedes-borne viruses are more urban than rural.

Response: We conducted a stratified analysis to address the potential biases and examined the magnitude and 95% CI in the associations of predictors (independent variables) with CHIKV, DENV, and ZIKV outcomes separately for urban and rural areas (lines: 231 – 233).

While the magnitude of measures of associations was slightly stronger for urban areas than for rural areas, the results show no differences in inferences. Regarding temperature, for example, the association with DENV outcome was relatively higher in urban than in rural areas. For population density, the association with DENV outcome was also slightly higher in urban than in rural areas. Inferences for urban in comparison with rural were similar for CHIKV and ZIKV (lines: 339 – 343).

Based on results in Figure S3-S23, the population density feature has limited impact for model predicting.

While this article focuses on statistical methods and validations, it lacks in biological understanding and interpretation. There is no mention of the value of ‘strength’ between indices when reported (how biologically relevant is having a unit difference?). It is fine to know that a given set of variables is associated with case counts, but I believe it would be more important to report what ranges of values are associated with observed patterns. What are the temperature values associated with clusters versus non-clusters? Based on the figures 2-4 (which, by the way, report mean without showing the 95%CI) the difference between clusters and non-clusters for a given factor appears to be minor. Omitting such information turns this paper into a mere statistical exercise. I would not be surprised that when the values are

presented, the differences are not only minor but also biased to the inclusion of rural areas (in cities, population density, temperature, may be less relevant than when including rural and even mountainous areas).

Response: The temperature features for cluster and non-cluster are same. Those are temperature related features in dataset.

In this study, we used aggregated data at the municipality level and the mean centre of each municipality was used in cluster detection.

We have added the following texts: “We found a positive association between mean temperature and CHIKV (Figures S6), DENV (Figures S7, S11, S12, S14, S15, S16, S19), and ZIKV transmission (Figure S21, S23), consistent with previous studies’ findings (8, 15, 24, 26, 27, 32). Our results are also consistent with other published findings that these mosquitoes can be infected with and can transmit all combinations of these viruses simultaneously within the observed temperature ranges in Mexico (49-51)” (Lines: 376 – 381).

In Table 2 and Table 4, we included standard error of prediction performance on CHIKV, DENV and ZIKV. From the results we can see the standard errors in above tables are relatively low which means our model’s performance is relative stable. In addition, we also added a figure S24 to indicate seasonality impact for CHIKV, DENV and ZIKV. From the plot we can see, seasonality has impact for DENV. For CHIKV and ZIKV, seasonality has marginally impact.

We also conducted a stratified analysis to address the potential biases (lines: 231 – 233).

Other (less crucial, yet important) issues are listed below:

The article is heavily focused on findings and analyses by the group, and ignores key knowledge and references in the subject (both in the background and discussion). There is plenty of evidence associating temperature with arbovirus transmission. In fact, temperature variability (range between min and max daily temperature) helps explain spatial variability in incidence at a finer scale than temperature alone. There are good descriptions of dengue distribution in Mexico at the municipal level and of co-circulation and overlap of the three Aedes-borne viruses at the city level which are also not mentioned here.

Response: We have updated the lit review and added the following texts in the introduction (lines: 88 – 96).

Smaller to larger dengue outbreaks were associated with increased temperature (23.8°C to 33.1°C) and the delayed effects could be predicted with a one-week lag (18). Mean (>27°C), minimum (>22°C) and maximum temperature (>38°C) was found to be the most favorable weather condition at a lag of one to three months in the tropical and subtropical climate zone, respectively (19, 20). Monthly mean rainfall showed a positive correlation with monthly dengue cases at a lag of one to three months (19-21). An increase of one cm of rainfall with a lag of two to three weeks was associated with 1.3 to 2.1% more dengue cases (22). Another study also suggested an increase of 1% in rainfall corresponded to an increase of 3.3% in the dengue cases (21).

Here we have added new references

Dzul-Manzanilla F, Correa-Morales F, Che-Mendoza A, Palacio-Vargas J, Sanchez-Tejeda G, Gonzalez-Roldan JF, Lopez-Gatell H, Flores-Suarez AE, Gomez-Dantes H, Coelho GE, da Silva Bezerra HS, Pavia-Ruz N, Lenhart A, Manrique-Saide P, Vazquez-Prokopec GM, 2021. Identifying urban hotspots of dengue, chikungunya, and Zika transmission in Mexico to support risk stratification efforts: a spatial analysis. *Lancet Planet Health* 5: e277-e285.

Santos JPC, Honorio NA, Barcellos C, Nobre AA. A Perspective on Inhabited Urban Space: Land Use and Occupation, Heat Islands, and Precarious Urbanization as Determinants of Territorial Receptivity to Dengue in the City of Rio De Janeiro. *Int J Environ Res Public Health*. 2020;17(18).

The reporting of results is very focused on the classification algorithms than in the key finding of spatio-temporal clusters (which leads to the classification). What was the duration of each spatio-temporal cluster? Was there any overlap in municipalities being clusters for 2 or more viruses? Why not trying ellipses rather than circles? [I believe the south cluster that has more area outside than inside the circle is a good example of bias].

Response: The duration of all significant clusters is for one specific time period. Here the disease cluster detection system is similar to disease hotspot (<https://desktop.arcgis.com/en/arcmap/10.3/tools/space-time-pattern-mining-toolbox/learnmoreemerging.htm>).

We put an enormous amount of time to run this clustering algorithm and used the entire study period to detect statistically significant spatiotemporal clusters and select the best and most meaningful findings for the classification algorithms.

We have updated figure 1. Yes, there are certain locations where 2 or more viruses were reported.

No limitations are mentioned in the discussion. For instance, relying on lab-confirmed cases would bias the data towards areas that do more testing, have more population, or have better infrastructure. Not acknowledging this is an issue.

Response: We agree and have added limitations: “The values of accuracy were higher than weighted accuracy and precision values, possibly due to the imbalanced scenario (the prevalence of people with infection is low and the data is sparse). The accuracy only considered the correct predicted cases and not which class the case come from. Therefore, if the dataset is imbalanced, we needed to add more evaluation metrics to evaluate the model performance. In our results, we used weighted accuracy, precision, recall, and F-score to evaluate our models. Here we used aggregated data at the municipality level. This study also used only the laboratory-confirmed cases and potentially lost many positive cases. The risk factors for ABDs transmission were also determined based on the passive surveillance system. In this study, we also used municipality-

level data and adjusted it for population density, seasonality, presence of *Ae. aegypti*, *Ae. albopictus*, rural/ urban classification, and altitude. Although the municipality level data has been widely used (47), it could be possible that some of the observed patterns are confounded by potential hidden factors in our data, as the many factors at the individual and household level may influence the distribution of *Ae. aegypti*. Identifying and addressing these hidden factors could be of great interest in future studies.” (lines: 430 – 443).

The distribution of ABDs infections is often driven by local spatiotemporal patterns influenced by fine-scale socio-economic, environmental, virological, and demographic factors (53-55). The current analysis at the municipality scale is too crude to capture many of these drivers of transmission heterogeneity (56). This implies a clear need for the development of a more integrated individual, and household level, with fine-scale time series data to understand the implications of these household patterns for targeted disease surveillance and vector control activities (lines: 445 – 451).

A figure with the incidence by municipality would have been very informative.

Response: We have added figure 1.

It is not clear whether the environmental and demographic conditions used for characterizing the separation between cluster and non-cluster areas involved the entire time series or only the period identified as Spatio-temporal cluster. The latter would be much more appropriate, and It seems it was not what has been done.

Response: To detect the geographical clusters, we used total counts of ABDs and population counts in each year, and geographical coordinates for each municipality. No environmental conditions were considered. Here only the period is identified as Spatio-temporal cluster.

Reviewer #3

In COMMSMED-21-0143, Dong et al. analyzed spatio-temporal data related to Dengue, Chikungunya, and Zika viruses in Mexico.

S1. The authors made good use of SaTScan to detect spatial clusters.

Response: Thank you very much.

S2. They also made good use of Pearson correlation coefficient and randomized dependence coefficient to analyze the socio-demographic and climatic impacts brought by these viruses and diseases.

Response: Thank you very much.

S3. They applied six existing machine learning models for clustering.

Response: Thank you very much.

S4. They also used SHapley Additive exPlanations for interpretation of results.

Response: Thank you very much.

S5. This showcases an application of machine learning in disease analytics.

Response: Thank you very much.

More details are needed for the Discussion section.

Response: We have substantially expanded the discussion section (lines: 376 - 381, 396 - 402, 404 - 443, 453 - 461).

What are the hypotheses?

Response: We have added the following to lines 112-114:

“The current study hypothesis is that there are geographic clusters of CHIKV, DENV, and ZIKV in Mexico. For some clusters, socio-economic features may have more impact on the prevalence of ABDs, whereas, in other clusters, climatic features have the greatest impact”.

What are the recommendations based on the results from this case study?

Response: We have added the following text (lines: 453 - 461)

“Future studies should be used to build predictive models to anticipate the time and location of ABDs outbreaks and determine the risk factors. Incorporating microclimate data, landscape ecology, urban environment into disease transmission models has the potential to yield more spatial precision and ecologically interpretable metrics of mosquito-borne disease transmission risk in urban landscapes (57-59). Further study of disease clusters concurrent with entomological data on *Aedes* distribution and human contact would also be beneficial. A better understanding of the drivers of ABDs transmission that consider local dynamics, should contribute to the design of a more effective mosquito control program, disease prevention, and promote public health in Mexico and other endemic countries.

References

1. Lubinda J, Trevino CJ, Walsh MR, Moore AJ, Hanafi-Bojd AA, Akgun S, et al. Environmental suitability for *Aedes aegypti* and *Aedes albopictus* and the spatial distribution of major arboviral infections in Mexico. *Parasite Epidemiol Control*. 2019;6:e00116.
2. Cheng J, Bambrick H, Yakob L, Devine G, Frentiu FD, Toan DTT, et al. Heatwaves and dengue outbreaks in Hanoi, Vietnam: New evidence on early warning. *PLoS Negl Trop Dis*. 2020;14(1):e0007997.
3. Akter R, Hu W, Gatton M, Bambrick H, Naish S, Tong S. Different responses of dengue to weather variability across climate zones in Queensland, Australia. *Environ Res*. 2020;184:109222.
4. Bal, S., Sodoudi, S. Modeling and prediction of dengue occurrences in Kolkata, India, based on climate factors. *Int J Biometeorol* 64, 1379–1391 (2020).
5. Polwiang S. The time series seasonal patterns of dengue fever and associated weather variables in Bangkok (2003-2017). *BMC Infect Dis*. 2020;20(1):208.
6. Hurtado-Diaz M, Riojas-Rodriguez H, Rothenberg SJ, Gomez-Dantes H, Cifuentes E. Short communication: impact of climate variability on the incidence of dengue in Mexico. *Trop Med Int Health*. 2007;12(11):1327-37.
7. Morgan J, Strode C, Salcedo-Sora JE. Climatic and socio-economic factors supporting the co-circulation of dengue, Zika and chikungunya in three different ecosystems in Colombia. *PLoS Negl Trop Dis*. 2021;15(3):e0009259.
8. Rodrigues NCP, Daumas RP, de Almeida AS, Dos Santos RS, Koster I, Rodrigues PP, et al. Risk factors for arbovirus infections in a low-income community of Rio de Janeiro, Brazil, 2015-2016. *PLoS One*. 2018;13(6):e0198357.
9. Whiteman A, Loaiza JR, Yee DA, Poh KC, Watkins AS, Lucas KJ, et al. Do socioeconomic factors drive *Aedes* mosquito vectors and their arboviral diseases? A systematic review of dengue, chikungunya, yellow fever, and Zika Virus. *One Health*. 2020;11:100188.
10. Charette M, Berrang-Ford L, Coomes O, Llanos-Cuentas EA, Carcamo C, Kulkarni M, et al. Dengue Incidence and Sociodemographic Conditions in Pucallpa, Peruvian Amazon: What Role for Modification of the Dengue-Temperature Relationship? *Am J Trop Med Hyg*. 2020;102(1):180-90.
11. Dzul-Manzanilla F, Correa-Morales F, Che-Mendoza A, Palacio-Vargas J, Sanchez-Tejeda G, Gonzalez-Roldan JF, et al. Identifying urban hotspots of dengue, chikungunya, and Zika transmission in Mexico to support risk stratification efforts: a spatial analysis. *Lancet Planet Health*. 2021;5(5):e277-e85.
12. Kinney RM, Huang CY, Whiteman MC, Bowen RA, Langevin SA, Miller BR, et al. Avian virulence and thermostable replication of the North American strain of West Nile virus. *J Gen Virol*. 2006;87(Pt 12):3611-22.
13. Ruckert C, Weger-Lucarelli J, Garcia-Luna SM, Young MC, Byas AD, Murrieta RA, et al. Impact of simultaneous exposure to arboviruses on infection and transmission by *Aedes aegypti* mosquitoes. *Nat Commun*. 2017;8:15412.
14. Bellone R, Failloux AB. The Role of Temperature in Shaping Mosquito-Borne Viruses Transmission. *Front Microbiol*. 2020;11:584846.
15. Salje H, Lessler J, Maljkovic Berry I, Melendrez MC, Endy T, Kalayanarooj S, et al. Dengue diversity across spatial and temporal scales: Local structure and the effect of host population size. *Science*. 2017;355(6331):1302-6.

16. Stoddard ST, Forshey BM, Morrison AC, Paz-Soldan VA, Vazquez-Prokopec GM, Astete H, et al. House-to-house human movement drives dengue virus transmission. *Proc Natl Acad Sci U S A*. 2013;110(3):994-9.
17. Bonifay T, Douine M, Bonnefoy C, Hurpeau B, Nacher M, Djossou F, et al. Poverty and Arbovirus Outbreaks: When Chikungunya Virus Hits More Precarious Populations Than Dengue Virus in French Guiana. *Open Forum Infect Di*. 2017;4(4).

Reviewers' comments:

Reviewer #1 (Remarks to the Author):

I acknowledge the authors have done a substantial revision, and put a lot of work into running varied types of analysis, but I still find this study to be very weak in its writing, description and justification of the methodology, evaluation and findings. Further, based on the discussion, the main findings seem to be mostly simple associations that could be inferred from the data directly.

I also think it is misleading to include co-circulation in the title, as the analysis does not provide a way to answer the question of co-circulation. The authors state "Our results are also consistent with other published findings that these mosquitoes can be infected with and can transmit all combinations of these viruses simultaneously within the observed temperature ranges in Mexico " but it is unclear what evidence they have for this, since the three diseases are all modeled independently, using reported data for each disease prevalence in humans.

Some specific concerns on the incomplete methods description:

The spatial analysis section does not make sense as written. The paragraphs are not linked, the objectives of each analysis are still unclear, and important information is still missing. For example:

Lines 213-224, I do not understand this entire paragraph. I believe it is focused on evaluating relative contribution of the impact of families of variables (SE vs climactic), but the specific variables are not listed. Just the themes. And the performance is described as based on combining 3 metrics. Do these metrics have units that can be combined? Does it make sense to average the value of multiple metrics together?

Line 226, what is the threshold and how was it determined, and what sensitivity analysis was conducted to justify the value chosen?

Line 231. potential biases of what?

Line 235: what was compared using these 6 methods? what is the baseline model for?

Line 241: how was cross validation used? within what experimental framework?

Reviewer #2 (Remarks to the Author):

The authors have addressed all my comments. Congratulations.

Reviewer #3 (Remarks to the Author):

In this revision COMMSMED-21-0143B-Z, Dong et al. analyzed spatio-temporal data related to Dengue, Chikungunya, and Zika viruses in Mexico. The authors addressed Reviewer 3's previous

concerns by adding more details to the Discussion section, and explicitly listing their hypotheses and recommendations.

Reviewer #1

I acknowledge the authors have done a substantial revision and put a lot of work into running varied types of analysis, but I still find this study to be very weak in its writing, description, and justification of the methodology, evaluation, and findings. Further, based on the discussion, the main findings seem to be mostly simple associations that could be inferred from the data directly.

Response: Thank you very much. We have also removed the sub-section of the analysis and brought all analyses under ‘data analysis’ and rearranged the analysis section to ensure the flow of the method section.

We have revised the conclusion in the abstract (lines: 53 – 58).

We also made some additional minor changes throughout the manuscript.

I also think it is misleading to include co-circulation in the title, as the analysis does not provide a way to answer the question of co-circulation. The authors state "Our results are also consistent with other published findings that these mosquitoes can be infected with and can transmit all combinations of these viruses simultaneously within the observed temperature ranges in Mexico " but it is unclear what evidence they have for this since the three diseases are all modeled independently, using reported data for each disease prevalence in humans.

Response: Thank you very much.

We have removed the word co-circulation from the title.

We have also modified texts throughout the manuscript so that it reads appropriately considering our lack of direct evidence to support the studies.

“While other published findings show that these mosquitoes can be infected with and can transmit all combinations of these viruses simultaneously within the observed temperature ranges in Mexico, our results indirectly support this evidence as evidence in the concurrent circulation of various arboviruses within the same population and geographic areas.” (lines: 386 – 390).

The spatial analysis section does not make sense as written. The paragraphs are not linked, the objectives of each analysis are still unclear, and important information is still missing. For example: Lines 220-231, I do not understand this entire paragraph. I believe it is focused on evaluating the relative contribution of the impact of families of variables (SE vs climactic), but the specific variables are not listed. Just the themes. And the performance is described as based on combining 3 metrics. Do these metrics have units that can be combined? Does it make sense to average the value of multiple metrics together?

Response: Thank you very much. We have reordered the texts (lines: 228 – 241).

The objectives of each analysis are clear now (lines: 195, 203).

We use figure 2A ~ figure 4B to illustrate the impact of a specific variable for *Aedes*-borne diseases (ABDs) (lines: 328 – 337).

These specific variables are included SE variables: illiteracy, population without health services, houses with a floor of dirt, houses without a toilet, houses without water pipelines, houses without sewage, houses without electricity, and climactic variables: mean max temperature, mean min temperature, mean temperature, mean mm/day, max mm/day, min mm/day, altitude (lines: 182 – 185, and 172 – 175 respectively).

In this study, we have used three metrics such as SHAP values, RDC, and Pearson coefficients to indicate the socio-economic and climate attributes' impact. For each metric, we aggregated the impact of all socio-economic variables and also aggregated the impact of all climate variables. Then, we compared the socio-economic and climate impact of three ABDs. Recall that SHAP values, RDC, and Pearson coefficients are used metrics for evaluating the variable impact for a specific model. In order to consider all three metrics, we designed two methods: 1) take average value 2) take majority vote value. The above two ways generally consider these three metrics and give more stable results (lines: 228 – 235).

Line 233 and 319, what is the threshold and how was it determined, and what sensitivity analysis was conducted to justify the value chosen?

Response: We chose the threshold based on a cross-validation experiment and the current threshold is set to 5. We select the threshold in our experiment which produces the best performance (line 248).

Please find recall (sensitivity, Tables 2 and 4) (lines 321).

Line 238. Potential biases of what?

Response: Thank you very much. We have revised the texts for more clarifications.

The data distributions may vary based on different ABDs. To ensure we consider each ABD separately and did not introduce data distribution bias, we conducted a stratified analysis to address the potential biases and examined the magnitude and 95% CI in the associations between predictors (independent variables) with CHIKV, DENV, and ZIKV outcomes separately for urban and rural areas (lines: 237 – 241).

Line 242: What was compared using these 6 methods? what is the baseline model for?

Response: Thank you very much. We have removed the sentence regarding the baseline methods and provided further explanations.

We also compared six commonly used prediction methods for the best model, such as XGBoost, decision tree, SVM with RBF kernel, KNN (K nearest neighbors) with five neighbors, random forest with six estimators, and neural network with 100 hidden layers. XGBoost is an implementation of gradient boosted decision trees designed for speed and performance (lines: 243 – 246).

Line 248: How was cross-validation used? within what experimental framework?

Response: We use 10 fold cross-validation procedure in our experiment (Table 1~ Table 4). We divide the dataset into 10 non-overlapping folds. Each of the 10 folds is given an opportunity to be used as a held back test set, whilst all other folds collectively are used as a training set. A total of 10 models are fit and evaluated on the 10 hold-out test sets and the mean performance is reported (lines: 253 – 256).

Reviewer #2

The authors have addressed all my comments. Congratulations.

Response: Thank you very much.

Reviewer #3

In this revision COMMSMED-21-0143B-Z, Dong et al. analyzed Spatio-temporal data related to Dengue, Chikungunya, and Zika viruses in Mexico. The authors addressed Reviewer 3's previous concerns by adding more details to the Discussion section, and explicitly listing their hypotheses and recommendations.

Response: Thank you very much.

REVIEWERS' COMMENTS:

Reviewer #1 (Remarks to the Author):

The authors have sufficiently addressed my previous comments.